# IntGrad MT: Enhancing LLMs' Machine Translation Capabilities with Sentence Interpolation Guided Gradual MT

## Abstract

Recent Large Language Models (LLMs) have demonstrated strong performance in translation without needing to be finetuned on additional parallel corpora. However, they still underperform for low-resource language pairs. Previous works have focused on mitigating this issue by leveraging relevant few-shot examples or external resources such as dictionaries or grammar books, making models heavily reliant on these nonparametric sources of information. In this paper, we propose a novel method named IntGrad MT that focuses on fully exploiting an LLM's inherent translation capability. IntGrad MT achieves this by constructing a chain of few-shot examples, each consisting of a source sentence and the model's own translation, that rise incrementally in difficulty. IntGrad MT employs two techniques: Sentence Interpolation, which generates a sequence of sentences that gradually change from an easy sentence to translate to a difficult one, and Gradual MT, which sequentially translates this chain using translations of earlier sentences as few-shot examples for the translation of subsequent ones. With this approach, we observe a substantial enhancement in the xCOMET scores of various LLMs for multiple languages, especially in low-resource languages such as Hindi(8.26), Swahili(7.10), Bengali(6.97) and Marathi(13.03). Our approach presents a practical way of enhancing LLMs' performance without extra training.

## 1 Introduction

Recent Large Language Models (LLMs) have shown strong performance in translation tasks without the need for fine-tuning on specific parallel datasets. Previous studies have demonstrated that LLMs' translation capabilities are reliable in most use cases, particularly when the source and target language are high-resource languages (Zhu et al., 2024; Robinson et al., 2023; Jiao et al., 2023). However, because LLMs require training on large corpora, they still face challenges when translating low-resource languages that are not sufficiently represented in the training corpora.(Stap & Araabi, 2023; Robinson et al., 2023; Enis & Hopkins, 2024).

Previous research has attempted to address these challenges by leveraging the in-context-learning capabilities of large language models (LLMs), particularly through the use of external knowledge such as few-shot examples or dictionaries during inference. However, relevant examples are not always guaranteed to be available, and constructing such external knowledge sources can be costly. A potential solution is to reduce reliance on external sources altogether.

In this paper, we examine whether we can improve LLMs' translation capabilities without relying on external knowledge. We aim to answer this question by considering a simple fact: the machine translation task can be defined as a mapping between two (sub)spaces for the source and target language. In the source language space, there are regions where the model performs well in translation and regions where it does not. The key idea is that if we can gradually expand the areas where the model performs well by feeding it neighboring examples, we can enhance its translation capabilities in the areas where it performs poorly.

We propose IntGrad MT to achieve this by connecting sentences from the regions where the LLM performs well in translation and regions where it does not. IntGrad MT consists of two key techniques: Sentence Interpolation and Gradual MT. Sentence Interpolation is a prompting technique

that generates a sequence of sentences gradually transitioning from one to another. Gradual MT is a technique in which the model iteratively translates a list of sentences, using its previous translations as few-shot examples for subsequent sentences. Through sentence interpolation, we first establish a pathway to the sentences in regions where the model performs poorly, then gradually expand the area where the model can excel, utilizing its in-context learning capabilities. The key concepts are illustrated in Figure 1.

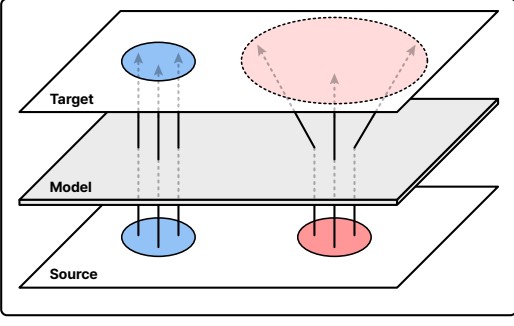 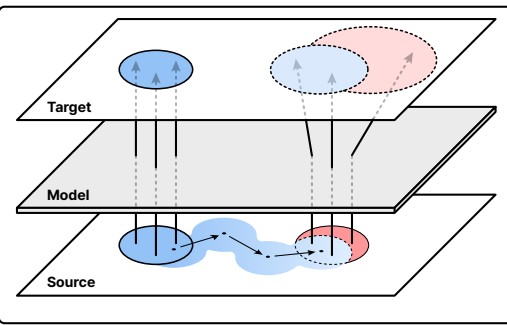

Normal          IntGrad MT

Figure 1: Figurative illustration of IntGrad MT. Machine translation task can be defined as a mapping between two (sub)spaces for the source and target language. In the source language space, there are regions where the model performs well in translation and regions where it does not. IntGrad MT expand the areas where the model performs well by feeding LLM with neighboring examples, eventually reaching the areas where it performs poorly.

We test the effectiveness of IntGrad MT by applying it to four different LLMs — GPT-3.5 Turbo, Mistral Nemo Instruct, Llama 3.1 70B Instruct, and Llama 3.1 8B Instruct — across seven target languages: German (De), Chinese (Zh), Hindi (Hi), Korean (Ko), Swahili (Sw), Marathi (Mr), and Bengali (Bn). Our results show consistently large performance gains, particularly in low-resource languages.

Additionally, we conduct an ablation study on English-to-Korean translation using ChatGPT to determine the optimal settings in relation to the following three questions:

- How should we select the start sentences for interpolation?
- How should we aggregate multiple translation results from different interpolation paths?
- To what sentences IntGradMT can be effective?

Our contributions are summarized as follows:

- We propose *IntGrad MT*, a novel method for machine translation that leverages two key methods: *Sentence Interpolation* and *Gradual MT*.
- We test the effectiveness of *IntGrad MT* with various LLMs and across different target languages.
- We conduct an ablation study to identify the optimal configuration.

## 2 RELATED WORK

### 2.1 ENHANCING LLMS' TRANSLATION CAPABILITIES WITHOUT FINE-TUNING

Modern LLMs show high capabilities in translation tasks in high- resource languages, but not in low-resource languages (Jiao et al., 2023; Stap & Araabi, 2023; Zhu et al., 2024; Enis & Hopkins, 2024). There are several works that have focused on enhancing LLMs' translation capabilities without additional fine-tuning. A primary strategy involves leveraging LLMs' ability to learn from

demonstrations or descriptions (Brown et al., 2020; Wei et al., 2022). Studies have explored selecting appropriate exemplars for few-shot learning and demonstrating linguistic knowledge (Agrawal et al., 2022; Vilar et al., 2023; Zhang et al., 2024), or augmenting LLMs with chains of multilingual dictionaries (Lu et al., 2024). Besides providing a demonstration or description, choosing the right temperature or prompting strategy has also been examined (Peng et al., 2023). Similar to previous research, our method aims to improve LLMs' MT capabilities without fine-tuning, focusing instead on eliciting the models' inherent capabilities with sentence interpolation and gradual MT rather than providing them with few-shot examples or external knowledge.

## 2.2 SELF-DEMONSTRATION

Manually generating appropriate exemplars for in-context learning can be resource-intensive. To address this challenge, previous studies have explored enabling models to generate their own few-shot examples for tasks such as classification(Lyu et al., 2023; Kim et al., 2022) or other reasoning tasks(Zhang et al., 2023; Li et al., 2024). Our work is aligned with these efforts, as it also focuses on generating the model's own few-shot examples. However, none of these approaches have tried to create examples such as ours, since our approach aims to produce the tailored example by gradually expanding the example set with similar yet distinct examples.

## 3 METHODS

We introduce the two steps that IntGrad MT consists of, Sentence Interpolation (§3.1) and Gradual MT (§3.2), and present how we combine them (§3.3).

### 3.1 SENTENCE INTERPOLATION

We propose sentence interpolation, which is a prompting technique asks model to create a list of sentences that gradually change from start sentence to end sentence. For example, following is the prompt that we use in our experiments:

> I will give you two sentences. Can you gradually change the first sentence to make it exactly the same as the second sentence? Just give me the sentences and don't provide additional comments.
>
> Sentence1: $\langle Sentence1 \rangle$
> Sentence2: $\langle Sentence2 \rangle$

The objective of this technique is to generate a list of sentences where each sentence is distinct, yet not excessively different from its adjacent sentences. By prompting the LLM to autonomously create the list, rather than mechanically altering sentences, we can obtain natural sentences that are suitable for use as in-context learning examples. We call this list of interpolated sentences the *interpolation path*. In practice, we utilized three few-shot examples from GPT-4(OpenAI et al., 2024) to control the output format. See Appendices A and B for these examples and sample interpolation paths.

### 3.2 GRADUAL MT

Gradual MT is a prompting technique that lets an LLM utilize its previous translations as a prompt. This approach sequentially processes a list of sentences, translating each one while using the previous translation results as few-shot examples for the current sentence. An illustration of the Gradual MT process is shown in Figure 2. Gradual MT can effectively guide LLMs in accurately translating unfamiliar sentences when combined with sentence interpolation. However, as Gradual MT is fundamentally a recursive process, it entails significant computational overhead. We discuss strategies to mitigate this cost in Section 6.3; *Path Truncation* and *Path Sampling*.

### 3.3 OVERALL METHOD

IntGrad MT combines sentence interpolation and gradual MT. The illustration of the algorithm can be seen in Figure 3.

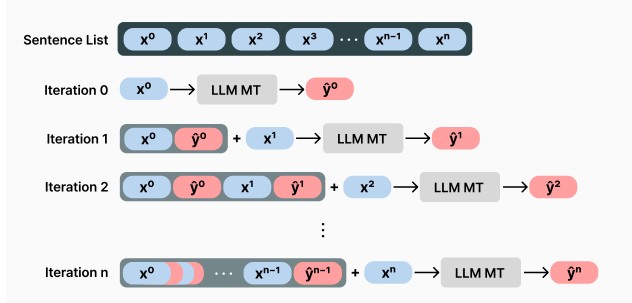

Figure 2: Illustration of Gradual MT. Gradual MT iteratively processes a list of sentences, translating each one while using the previous translation results as few-shot examples for the current sentence.

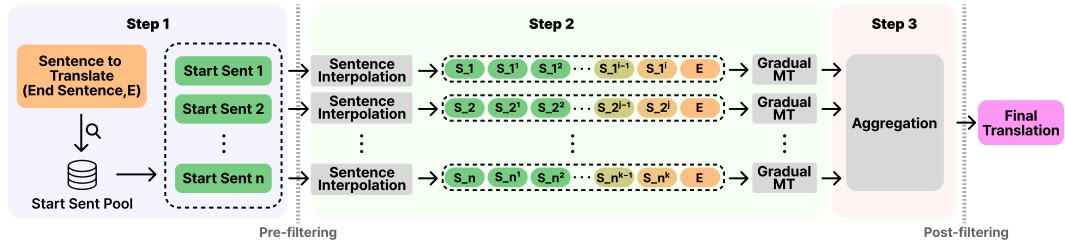

Figure 3: Illustration of the IntGrad MT algorithm. IntGrad MT integrates Sentence Interpolation with Gradual MT. In Step 1, $n$ start sentences are selected from a pre-defined start sentence pool. In Step 2, these start sentences are interpolated toward the end sentence, creating $n$ individual interpolation paths. Each path is then processed through Gradual MT, generating translation results for every sentence along the path. In Step 3, the MT results from all interpolation paths are aggregated to produce a single output translation. Optional pre- and post-filtering steps can be applied between Steps 1 and 2, and after aggregation, to refine the sentences on which IntGrad MT is applied.

**Step 0: Start Sentence Pool Creation.** Before applying IntGrad MT, the *start sentence pool* must be created before. This pool consists of sentences that the LLM can already translate accurately. It is crucial to ensure that the source sentences included in this pool have been tested and verified, confirming the LLM can generate high-quality translations in a zero-shot setting. We refer to the source sentences in the start sentence pool as *start sentences*.

**Step 1: Start Sentence Selection** IntGrad MT begins with selecting $n$ start sentences from the start sentence pool by calculating the similarity with the source sentence that the LLM is trying to translate, which we call an *end sentence*. Measuring similarity between sentences can be done in various ways. In this paper, we utilized SBERT similarity(Reimers & Gurevych, 2019) as a primary metric. Details about start sentence selection strategy can be found in Section 6.1.1.

**Step 2: Sentence Interpolation & Gradual MT.** After selecting $n$ start sentences, sentence interpolation is performed between each start sentence and the corresponding end sentence, creating $n$ individual *interpolation paths*. The paths are then processed through the Gradual MT, generating translation results for each sentence.

**Step 3: MT Results Aggregation.** After generating a list of translation, we proceed to aggregate the final translation from each path. If a single start sentence was selected in Step 1, this step is skipped. There are several methods for aggregating translations; in our approach, we input all the translation results into the LLM once again as few-shot examples to generate the final translation. A detailed explanation of this aggregation strategy can be found in Section 6.1.2. After aggregation, we obtain the final translation, which we refer to as the *output*.

**Pre- & Post-filtering**  IntGrad MT can be applied to all end sentences; however, it is often more effective when used selectively for two reasons. First, determining which sentences will undergo IntGrad MT before execution can significantly reduce computational costs. Second, even after applying IntGrad MT, the output can be discarded if its translation quality has degraded. We explored these two possibilities, referred to as *Pre-filtering* and *Post-filtering*, in our ablation study (§6.1.3).

## 4 EXPERIMENT

### 4.1 SETUP

**Models.**  For **translation** we use four different LLMs: ChatGPT(GPT-3.5-Turbo-0125)[1], Mistral Nemo(Mistral-Nemo-Instruct-2407)[2] and two different sized Llama 3.1 Instruct models(Llama-3.1-70B-Instruct, Llama-3.1-8B-Instruct)(Dubey et al., 2024). ChatGPT is accessed via OpenAI's API, and the others are run locally. For **sentence interpolation**, we employ Qwen2-72b-Instruct (Yang et al., 2024) with quantization. See Appendices D and E for settings for translation and interpolation. For **pre- and post-filtering**, we utilize a reference-free QE model CometKiwi (Rei et al., 2023) to avoid peeking at the gold translations. CometKiwi produces a DA score, which rates translation quality on a scale from 0 to 100, normalized to a range of 0 to 1. Lastly, we used all-mpnet-base-v2[3] for **SBERT sismilarity** calculation.

**Target Languages.**  We fixed English as the source language. The target languages tested in the experiments are German (De), Chinese (Zh), Korean (Ko), Hindi (Hi), Swahili (Sw), Bengali (Bn), and Marathi (Mr). Based on Joshi et al. (2020)'s 6 scale taxonomy of language resource level, we classify German and Chinese as high resource languages, Korean and Hindi as mid resource, and the rest as low resource. For the Llama 3.1 models, we conduct tests only on German and Hindi, as those models do not support other languages. Mistral Nemo Instruct does not officially support Ko, Hi, Sw, Bn, and Mr, but we conduct experiments on those languages nevertheless, as it has some capability to generate them.

**Dataset.**  We use the FLORES-200 benchmark dataset(Team et al., 2022) for validation and evaluation. We first utilize the dev split of the dataset to create start sentence pool. During evaluation, to test the effect of the pre-filtering strategy, we selected 10% of the test portion of the dataset to set the DA score threshold. The remaining 90% is used to assess the overall performance on the dataset.

**Start Sentence Pool Creation.**  The initial sentence pool is created by translating source sentences from the dev split of the FLORES-200 dataset using a zero-shot approach. Each source sentence is translated five times and evaluated with xCOMET (Guerreiro et al., 2024). xCOMET predicts DA score normalized to a range of 0 to 1. The most frequently occurring translation is selected as the *representative translation*. If no translation is repeated, the one with a score closest to the average is chosen. After selecting each representative translation, the top 100 translation pairs with the highest DA scores are selected.

**Baselines.**  We compare zero-shot MT results with 15-shot and 50-shot MT using source sentences from the start sentence pool and their gold translations. Additionally, we benchmark against TowerInstruct 13B (Alves et al., 2024) and NLLB-200-3.3B[4] for broader comparison.

**Evaluation.**  We use the DA score, as evaluated by xCOMET, as the primary metric for our evaluation, scaling it by a factor of 100 for improved readability. Additionally, we employ MetricX (Juraska et al., 2023) to compute an MQM score, which assesses translation errors on a scale from 0 to 25, where lower scores indicate higher quality. We also evaluate the DA score using CometKiwi and BLEURT (Pu et al., 2021). Finally, we report the BLEU score. Computational costs associated with executing IntGrad MT are detailed in Appendix C.

---

[1]https://platform.openai.com/docs/models/gpt-3-5-turbo

[2]https://huggingface.co/mistralai/Mistral-Nemo-Instruct-2407/

[3]https://huggingface.co/sentence-transformers/all-mpnet-base-v2

[4]https://huggingface.co/facebook/nllb-200-3.3B

## 5 RESULTS

We first conducted various ablation settings for English-Korean translation using ChatGPT to identify the optimal configuration such as start selection(step 1) and result aggregation(step 3). Subsequently, this configuration was applied to other experiments with different LLMs and languages. A detailed explanation of the ablation study is provided in section §6.1. In this section, we report xCOMET results of IntGrad MT on various LLMs and target languages in Table 1. Results based on other metrics can be found at Appendix F.

Table 1: xCOMET scores of IntGrad MT with the best ablation setting for different LLMs and target languages. The strategy used consists of selecting the starting sentences by sorting with SBERT similarity and tree edit distance(Sort(S-T)), aggregating MT results by feeding all gradual MT results into the LLM(Prompt), and selecting the output MT only if its CometKiwi score is higher than the zero-shot translation(Post-filtering). The results of applying pre- and post-filtering together are presented below the main results as supplementary information. Zero-shot, 15-shot and 50-shot are the baselines, with the examples selected using the same starting sentence selection method as IntGrad MT. Scores are multiplied by 100 for readability, and the two highest scores for each MT model and language are highlighted in bold and underlined. Llama translation results for languages other than De and Hi are not available, as Llama does not support those languages.

| MT Model | Method | High Resource | | Mid Resource | | Low Resource | | |
| --- | --- | DE | ZH | HI | KO | SW | BN | MR |
| | 0 shot | 97.63 | 91.30 | 71.89 | 89.48 | 81.23 | 68.73 | 44.53 |
| | 15 shot | 98.01 | 92.16 | 73.13 | 90.73 | 81.59 | 69.70 | 45.54 |
| GPT 3.5 | 50 shot | 97.99 | 91.95 | 72.85 | 90.93 | 82.10 | 67.72 | 44.84 |
| | Intgrad MT_Post | **98.04** | **92.42** | **77.54** | **92.54** | **84.03** | **75.60** | **53.57** |
| | Intgrad MT_Pre & Post | 97.95 | 91.54 | 77.27 | 92.08 | 83.73 | 74.78 | 52.31 |
| | 0 shot | 97.33 | - | 79.41 | - | - | - | - |
| | 15 shot | 97.65 | - | 77.95 | - | - | - | - |
| Llama 70b | 50 shot | 97.41 | - | 72.68 | - | - | - | - |
| | Intgrad MT_Post | **97.98** | - | **84.45** | - | - | - | - |
| | Intgrad MT_Pre & Post | 97.33 | - | 84.35 | - | - | - | - |
| | 0 shot | 94.99 | - | 69.93 | - | - | - | - |
| | 15 shot | 96.26 | - | 73.70 | - | - | - | - |
| Llama 8b | 50 shot | **96.42** | - | 73.36 | - | - | - | - |
| | Intgrad MT_Post | 96.42 | - | **78.18** | - | - | - | - |
| | Intgrad MT_Pre & Post | 95.37 | - | 77.33 | - | - | - | - |
| | 0 shot | 96.70 | 88.01 | 66.78 | 81.43 | 38.97 | 71.89 | 43.99 |
| | 15 shot | 97.68 | 90.94 | 69.54 | 88.71 | 42.79 | 73.60 | 52.26 |
| Mistral Nemo | 50 shot | 97.72 | 90.92 | 69.35 | 88.29 | 42.53 | 73.80 | 52.04 |
| | Intgrad MT_Post | **97.88** | **91.04** | **74.19** | **89.38** | **46.08** | **78.86** | **57.02** |
| | Intgrad MT_Pre & Post | 97.84 | 91.04 | 73.47 | 89.31 | 44.53 | 78.67 | 56.79 |
| NLLB | - | 96.21 | 67.88 | 81.00 | 82.20 | 77.17 | 82.70 | 71.83 |
| TowerInstruct | - | 97.69 | 89.89 | - | 91.29 | - | - | - |

As shown in Table 1, IntGrad MT outperforms the baselines in all models for all languages except for German with Llama-3.1-8b. This indicates that LLMs possess hidden multilingual and linguistic knowledge that can be exploited for machine translation with the correct techniques. Compared to the zero-shot baseline, IntGrad MT increases the xCOMET score by a maximum of 3.03, 8.26, and 13.03 points in high, mid, and low resource languages, respectively. It incurs the largest improvements for low- and mid- resource languages, suggesting that it holds the potential to improve the quality of translation for these languages that LLMs are currently unable to translate well. This supports the advantage of our method in formulating tailored prompts using the example translation pairs over simply listing them. The results are consistent with other metrics too(Table 9, 8 & 10), except for BLEU (Table 11), which showed only marginal improvement in comparison.

For a more balanced examination of how much gains IntGrad MT brings relative to the original performance, we compared the error reduction rate across three different metrics, xCOMET, CometKiwi, and MetricX, in Figure 4. Since the DA score reflects how 'good' the translation is, the error for the DA score was calculated by subtracting the score from 100 ($100 - DA$). The MQM score from MetricX was used directly, as it inherently represents the error. IntGrad MT consistently reduces errors in all metrics for all languages with only one exception. Compared to Table 1 the improvements to the high resource languages are more clearly visible.

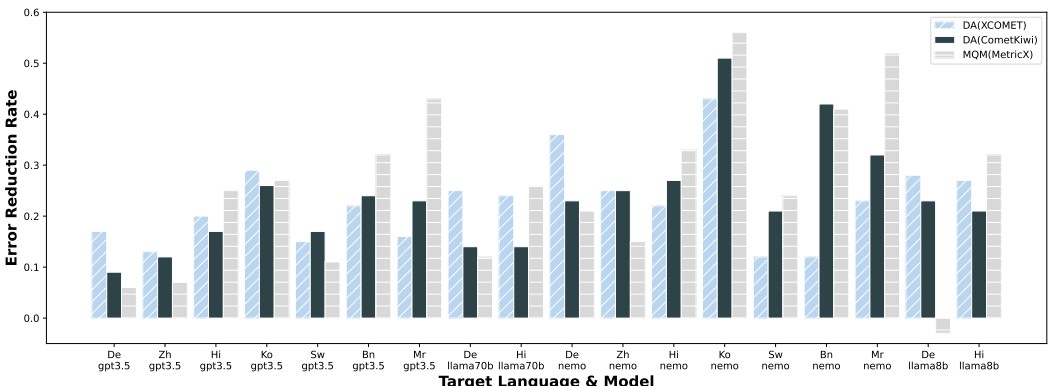

Figure 4: Error reduction rate for each target language and model, calculated as the ratio of reduced error relative to the error of zero-shot translation results. Error scores from xCOMET and CometKiwi were obtained by subtracting their values from 100, while MetricX's score was used directly.

# 6 ANALYSIS

## 6.1 ABLATION STUDY

To find the optimal combination of strategies for the IntGrad MT, we conducted an ablation study on En-Ko translation task using ChatGPT in terms of four different dimensions: Start sentence selection, Number of start sentences, MT result aggregation, and Filtering. In this section, we analyze the effects of each strategy by averaging the QE scores. Full results with all different combination of ablation settings can be found in Appendix G.

### 6.1.1 START SENTENCE SELECTION STRATEGY

Based on the intuition that start sentences that are similar to the source sentence will be helpful, we utilized three different metrics — SBERT similarity(Reimers & Gurevych, 2019), Levenshtein distance(Levenshtein & others, 1966), and Tree edit distance(Zhang & Shasha, 1989) — to calculate similarity between sentences, and combined them in theee different ways. The first way is *Sort*, which sorts sentences by multiple metrics with varying priorities. The second way, *Filter*, initially selects the top 10 sentences based on SBERT similarity, then sorts the selection using the other metrics. The third way, *Tops*, picks the top sentence based on the highest similarity scores from each metric. As shown in Table 2, the selection strategies that produced the highest scores differed across aggregation methods. We chose sorting by SBERT similarity and then tree edit distance (Sort(S-T)), as it yielded the highest average scores.

### 6.1.2 NUMBER OF START SENTENCES & MT AGGREGATION STRATEGY

When selecting the start sentences, we need to decide whether to use more than one start sentence. If we choose more than one, we must aggregate each translation result generated by Gradual MT. We investigated two distinct strategies for aggregating the results of Gradual MT. The first method, referred to as Polling, selects the MT result with the highest number of duplicates, drawing inspiration from prior research on self-consistency (Wang et al., 2023). If no duplicates are found, a result is selected randomly. The second method, Prompting, involves feeding all Gradual MT results into the LLM as few-shot examples to generate the final MT output. As shown in Table 2, the prompting strategy outperforms polling by 0.49 points. Polling is even worse than using a single start sentence.

### 6.1.3 FILTERING STRATEGY.

We tested three strategies for filtering. The first strategy, 'Pre-filtering', aims to minimize costs by evaluating zero-shot translation results with a QE model and applying IntGrad MT only when the QE

Table 2: Averaged scores with different start selection strategies, start sentence numbers and aggregation strategies applied to EN-KO translation task. 'Sort', 'Filter', and 'Tops' denote the start sentence selection methods. Each letter in parentheses represents a similarity metric, with the order indicating the priority of these metrics. 'L', 'T' and 'S' stands for Levenshtein distance, Tree edit distance, and SBERT similarity, respectively. The highest average values for each axis are highlighted in bold. The highest values for each start sentence number and aggregation strategy are highlighted in underline.

| Start Selection Strategy | Start Sents.Num & Aggregation Strategy | | | Average |
|---|---|---|---|---|
| | 1 (n/a) | 3 Poll | 3 Prompt | |
| Filter(T-L) | 91.29 | 91.22 | 91.72 | 91.41 |
| Filter(L-T) | 91.67 | 91.23 | 91.59 | 91.50 |
| Sort(L-S) | 91.35 | 91.22 | 91.57 | 91.38 |
| Sort(T-S) | 90.95 | 91.09 | 91.65 | 91.23 |
| Sort(L-T-S) | 91.19 | 91.21 | 91.50 | 91.30 |
| Sort(T-L-S) | 91.19 | 91.00 | 91.93 | 91.37 |
| Sort(S-T) | 91.50 | 91.39 | 91.83 | **91.57** |
| Tops | - | 91.23 | 91.69 | 91.46 |
| **Average** | 91.31 | 91.20 | **91.69** | |

score falls below a certain threshold. The second strategy, 'Post-filtering', prioritizes maximizing performance by applying IntGrad MT first and using its output only if the QE score exceeds that of the zero-shot translation. The third strategy, 'Pre- & Post-filtering' combines the first two: applying IntGrad MT when the zero-shot translation's QE score is below a threshold, and only if IntGrad MT's score is higher. We employed CometKiwi, a reference-free QE model, to implement these strategies. To compare them, we analyzed xCOMET scores and score changes of selected outputs, applying the optimal strategies for start selection ('Sort(S-T)') and aggregation ('Prompting'). As shown in Table 3, all strategies improved overall performance, with 'Post-filtering' achieving a notable gain of over 1 point in QE scores compared to zero-shot MT. The results also indicate that 'Pre- & Post-filtering' reduces interpolation by more than half while maintaining comparable performance to 'Post-filtering', offering an effective compromise between computational efficiency and translation quality. Results with every combination of ablation strategies are shown in Table 13.

Table 3: Average xCOMET scores and score changes of selected outputs when applying the optimal strategies for start selection and aggregation in En-Ko translation. 'Score change' is calculated only for the adopted outputs. 'All' selects every output. 'Pre-filtering', denoted as 'Pre', applies zero-shot translation first and uses IntGrad MT only when the CometKiwi score is below a threshold. 'Post-filtering', denoted as 'Post', selects outputs only if they outperform zero-shot translations. 'Pre- & Post- filtering', denoted as 'Pre & Post' combines these two strategies. We also report the number of end sentences for which interpolation and Gradual MT is executed ('No. of Interpolated End Sents') and the number of end sentences for which the IntGrad MT output is selected over zero-shot translation ('No. of Selected Outputs'). The results show that 'Pre & Post' reduces the number of interpolation by more than half while maintaining nearly the same translation performance.

| Filtering Strategy | Avg. Score | Avg. Score Change of Selected Outputs | No. of Interpolated End Sents(%) | No. of Selected Outputs (%) |
|---|---|---|---|---|
| Zeroshot | 89.48 | - | - | - |
| All | 91.22 | 1.74 | 911 (100%) | 911 (100%) |
| Pre | 91.50 | 4.49 | 410 (45%) | 410 (45%) |
| Post | **92.54** | 5.61 | 911 (100%) | 497 (54%) |
| Pre & Post | 92.08 | **8.50** | 410 (45%) | 279 (31%) |

Based on the results of the ablation study, we concluded that the optimal combination of strategies is as follows: for **Start Sentence Selection**, sorting with SBERT similarity followed by tree edit distance proved most effective(Sort(S-T); for the **Number of Start Sentences**, using 3 was optimal;

for **Aggregation**, the best method was prompting; and for **Filtering**, both 'Post-filtering' and 'Pre-& Post-filtering' were optimal.

## 6.2 SENTENCE INTERPOLATION ANALYSIS

We conducted an analysis of the interpolated sentences used in the ablation study to verify whether they effectively interpolate between the start and end sentences. The total number of sentences in interpolated paths is 12,729, and the average length of interpolation path is 7.31.

**Sentence Interpolation Error Rate.**   We defined that a situation where the first and last sentences in the interpolation path are not exactly the same as the start and end sentences as an *error*, and executed zero-shot translation for that end sentence. Sentence interpolation was successfully executed without error in 93.68% of cases.

**Progresses of Interpolation.**   We examined whether LLMs genuinely interpolate between start and end sentences or simply generate random sentences. To assess this, we use SBERT to embed the interpolated sentences and calculate their Euclidean distances from the end sentence. If these distances generally decrease, it indicates successful interpolation. To mesure this, we defined *progress* as progress $= d_{n-1,e} - d_{n,e}$ where $d_{n,e}$ denotes the euclidian distance between $n^{\text{th}}$ interpolated sentence in each interpolation path and end sentence.

As shown in Figure 5, the progress of interpolated sentences is generally positive, indicating that the interpolation effectively bridges the two sentences. The average progress for all interpolated sentences is 0.14, with a standard deviation of 0.23.

We also conducted a qualitative analysis to identify patterns in interpolation. First, we sampled 100 interpolation paths for En-Ko translation containing more than three sentences and obtained sentence embeddings using SBERT. Next, we plotted each interpolation path on a 2D plane using Principal Component Analysis (PCA) and observed the patterns. After the analysis, we confirmed that sentence interpolation typically shifts the start sentences toward the end sentences, despite the variety of patterns (Arc, Triangle, Zig-Zag, and Leap). Detailed explanations of each pattern can be found in Appendix I.

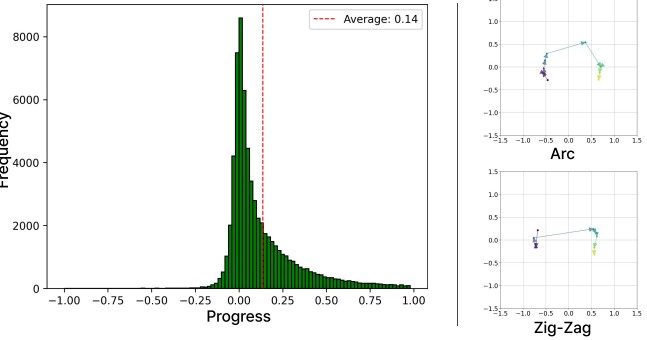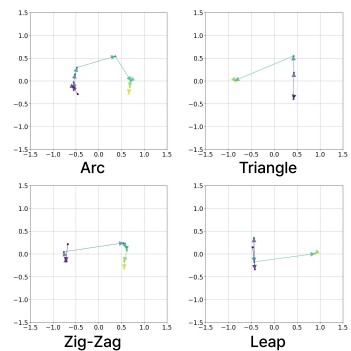

Figure 5: Distribution of progress in sentence interpolation (left) and 2D scatter plots showing four major patterns of interpolation paths (right). Interpolation paths were sampled from the En-Ko ablation study. Progress indicates how much each interpolated sentence moves closer to the target sentence. Scatter plots are projected from SBERT embeddings onto a 2D plane using PCA. The X and Y axes of each plot represent the first and second principal components, respectively. Arrows in each plot show the trajectory of sentence shifts from the start sentence (blue-colored dot) to the end sentence (yellow-colored dot).

## 6.3 STRATEGIES TO SAVE COMPUTATIONAL COST OF GRADUAL MT

To reduce the computational overhead induced by the recursive translations of Gradual MT, we further explored two strategies: *Path Truncation* and *Path Sampling*. **Path Truncation** uses a fixed

Table 4: xCOMET scores of IntGrad MT with different strategies to save computational cost of gradual MT. 'Default' denotes the strategy which uses every sentences from interpolation path. 'Path truncation' denotes the strategy which uses three recent translations for gradual MT. 'Path sampling' denotes the strategy which uses start, middle and end sentence from the interpolation path for gradual MT.

| SETTING | DE | ZH | HI | KO | SW | BN | MR |
|---|---|---|---|---|---|---|---|
| **15 shot** | 98.01 | 92.16 | 73.13 | 90.73 | 81.59 | 69.70 | 45.54 |
| **Default (1 start)** | 97.85 | 91.36 | 73.89 | 90.68 | 82.18 | 69.83 | 47.36 |
| +Pre | 97.87 | 91.89 | 74.80 | 91.30 | 82.43 | 71.84 | 48.53 |
| +Post | 98.02 | 92.51 | 76.44 | 92.29 | 83.96 | 74.16 | 51.70 |
| +Pre&Post | 97.93 | 92.04 | 76.32 | 91.70 | 83.93 | 73.66 | 50.78 |
| **Path Truncation** | 97.87 | 91.35 | 74.42 | 90.27 | 82.12 | 69.88 | 46.96 |
| +Pre | 97.91 | 91.87 | 74.94 | 90.77 | 82.42 | 71.61 | 48.57 |
| +Post | 98.06 | **92.60** | 77.07 | 91.98 | 83.87 | 73.85 | 51.55 |
| +Pre&Post | 97.96 | 92.04 | 76.84 | 90.93 | 83.86 | 73.18 | 50.94 |
| **Path Sampling** | 97.97 | 91.13 | 78.42 | 92.72 | 83.73 | 74.49 | 52.84 |
| +Pre. | 98.00 | 91.30 | 79.02 | 92.99 | 84.39 | 76.17 | 55.35 |
| +Post | **98.18** | 92.19 | **80.62** | **94.16** | **85.77** | **78.75** | **57.95** |
| +Pre&Post | 98.13 | 91.35 | 80.48 | 93.23 | 84.77 | 77.64 | 57.07 |

number of recent examples during Gradual MT. **Path Sampling** selects a fixed number of sentences from the interpolation path for Gradual MT. Path Truncation reduces the number of tokens required, while Path Sampling controls the number of iterations, thereby reducing the token count as well. We tested these two methods in a scenario using a single start sentence with ChatGPT as the translation model. For Path Truncation, we used the three most recent examples from the iteration. For Path Sampling, we extracted three sentences from the path: start, middle, and end sentences. All other settings matched the optimal configuration determined in our ablation study. As shown in Table 4, Path Truncation resulted in a slight performance degradation, indicating that it can serve as a viable option when computational cost savings are critical. Interestingly, Path Sampling outperformed the default setting, particularly for low-resource languages. This suggests that Path Sampling may mitigate potential noise introduced by a long path. BLEU scores can be found in Appendix H.

# 7 CONCLUSION

In this paper, we proposed IntGrad MT, a novel method to enhance the machine translation capabilities of various LLMs. IntGrad MT leverages sentence interpolation to guide models, eliciting stronger translation performance. Experimental results across various models and languages demonstrate that our approach consistently improves translation quality, particularly in low-resource languages, achieving meaningful gains in performance metrics. Our approach is practical in that it does not require extra training and does not conflict with previous methods that utilize other kinds of prompting techniques.

# 8 LIMITATIONS

Despite its success, IntGrad MT introduces significant computational overhead, particularly in scenarios involving multiple start sentences or large-scale models. Future work could focus on optimizing computational efficiency (as we did in Section 6.3) or extending the approach to specialized domains and additional low-resource language pairs. Moreover, since sentence interpolation did not perform well in languages other than English, even with GPT-4 (OpenAI et al., 2024), we had to limit our focus to cases where English was the source language. Exploring better prompting techniques to interpolate non-English sentences is a potential future research direction.

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

## A  FEW-SHOT EXAMPLES FOR SENTENCE INTERPOLATION

In this study, we gave three examples to the interpolation model. We arbitrarily picked start and end sentences from the New York Times news articles.

### A.1  EXAMPLE 1

**Sentence 1:** Surprisingly often, animals show up uninvited at sporting events. Sometimes, it gets a little weird.

**Sentence 2:** D. Cohen tried to parry attacks on his honesty as Todd Blanche, Donald J. Trump's lawyer, tried to destroy his credibility with jurors.

**Interpolation path**

1. Surprisingly often, animals show up uninvited at sporting events. Sometimes, it gets a little weird.

2. Often, unexpected events occur during live events, and sometimes these can get quite weird.

3. At public events, such as sports or trials, unexpected guests often cause disturbances, leading to weird situations.

4. In public spectacles, like sports or courtrooms, unexpected participants can disrupt the normal proceedings in strange ways.

5. During public hearings, like in court, surprising accusations and defenses can emerge, often causing odd disruptions.

6. In courtroom battles, unexpected arguments and witnesses can often turn the proceedings weird as each side tries to undermine the other.

7. In legal battles, lawyers frequently confront each other with surprising tactics to challenge credibility, which can make the proceedings seem strange.

8. Michael D. Cohen, during his trial, encountered surprising tactics by Todd Blanche, Donald J. Trump's lawyer, aiming to disrupt his credibility in unusual ways.

9. Michael D. Cohen tried to parry attacks on his honesty as Todd Blanche, Donald J. Trump's lawyer, tried to destroy his credibility with jurors.

## A.2 EXAMPLE 2

**Sentence 1:** Georgia's Parliament overrode a presidential veto to give final approval on Tuesday evening to a contentious bill that has plunged the country into a political crisis and threatened to derail the pro-Western aspirations of many Georgians in favor of closer ties with Russia.

**Sentence 2:** At that time, the buildings were abandoned and inaccessible to the public, and the famous ferry that shuttled immigrants from the island to Manhattan sank in a storm in 1968.

**Interpolation Path**

1. Georgia's Parliament overrode a presidential veto to give final approval on Tuesday evening to a contentious bill that has plunged the country into a political crisis and threatened to derail the pro-Western aspirations of many Georgians in favor of closer ties with Russia.

2. The decision made by Georgia's Parliament on Tuesday evening, to override a presidential veto and approve a controversial bill, has sparked a political crisis in the country and raised concerns about the impact on the country's pro-Western stance and relations with Russia.

3. The controversial bill approved by Georgia's Parliament on Tuesday, despite a presidential veto, has caused a political crisis and stirred debates about the country's Western alignment and its relation with Russia.

4. The unilateral move by Georgia's Parliament on Tuesday to push through a divisive bill despite opposition from the president has led to a political crisis and raised concerns about the nation's Western affiliations and ties with Russia.

5. The recent decision by Georgia's Parliament to defy the presidential veto and pass a controversial bill has created a political turmoil, questioning the country's Western orientation and relationship with Russia.

6. Georgia's Parliament's decision to override the presidential veto and approve the contentious bill has caused a political crisis in the country, prompting discussions on the impact on its Western ties and relations with Russia.

7. Georgia's Parliament's act of rejecting the presidential veto and endorsing the controversial bill on Tuesday has resulted in a political crisis, with implications for the country's pro-Western stance and connections with Russia.

8. Georgia's Parliament moved forward with finalizing a contentious bill on Tuesday, overriding the presidential veto, which has triggered a political crisis and raised concerns about the nation's Western aspirations and relationship with Russia.

9. At that time, the buildings were abandoned and inaccessible to the public, and the famous ferry that shuttled immigrants from the island to Manhattan sank in a storm in 1968.

## A.3 EXAMPLE 3

**Sentence 1:** The cake has become one of Jamaica's most popular desserts, often found in various stores.

**Sentence 2:** Some plans might cover only generic versions of certain contraceptives, but patients are still entitled to coverage of a specific product that their providers deem medically necessary.

**Interpolation Path**

1. The cake has become one of Jamaica's most popular desserts, often found in various stores.

2. Some plans might only cover basic versions of popular desserts like the cake, while others are more comprehensive.

3. In many cases, plans might cover only generic versions of popular products, but specific varieties might still be available.

4. Some plans might cover only generic versions of certain products, with exceptions allowed for specific needs.

5. Some plans might cover only generic versions of certain healthcare products, but specific options are still accessible if necessary.

6. Some plans might cover only generic versions of certain contraceptives, but patients can still get specific types if deemed necessary.

7. Some plans might cover only generic versions of certain contraceptives, but patients are still entitled to coverage of a specific product that their providers deem medically necessary.

## B  SAMPLE SENTENCE INTERPOLATION PATH

Below is a sample of the interpolation path generated by Qwen2-72b-Intstruct during the experiment.

### B.1  SAMPLE 1

**Prompt**
Sentence1: I am happy that there are people willing to support me.
Sentence2: "We now have 4-month-old mice that are non-diabetic that used to be diabetic," he added.

**Output**

1. I am happy that there are people willing to support me.

2. It's good to know that support is available when needed.

3. It's reassuring to see support being provided in various situations.

4. He mentioned seeing positive changes in different scenarios.

5. "We have seen positive transformations in our subjects," he added.

6. "We now have improved conditions in our test subjects," he added.

7. "We now have 4-month-old mice that are non-diabetic that used to be diabetic," he added.

## B.2 SAMPLE 2

**Prompt**
Sentence1: The death toll is at least 15, a figure which is expected to rise.
Sentence2: The moisture on your hands will react with the outer layers, which will feel funny and form a sort of shell.

**Output**

1. The death toll is at least 15, a figure which is expected to rise.

2. The number affected is at least 15, an amount which could increase due to underlying factors.

3. An initial count shows at least 15 impacted, with this figure potentially growing as outer layers are peeled back.

4. A preliminary assessment reveals at least 15 affected individuals, with the outer layers revealing more as they are examined.

5. The moisture on your hands will interact with the outer layers, which may reveal more than initially thought.

6. The moisture on your hands will react with the outer layers, which will feel peculiar and begin to peel away.

7. The moisture on your hands will react with the outer layers, which will feel funny and form a sort of shell.

# C COMPUTATIONAL COST

Table 5: The computational cost (execution time and memory) for IntGrad. The statistics are calculated translating 10 randomly sampled sentences from the FLORES test set into Hindi using Llama-3.1-8b. *Path sampling* denotes the gradual MT operation using only the first, middle, and last steps of the interpolation path. The pre-filtering and post-filtering steps run the QE model using batches of data. We used a batch size of 8, and running one batch took 49.77 seconds. Since the time of running N sentences can be estimated at 49.77 * (N/8), we denote the time per sentence as 6.22(s).

| Step | Time per sentence (s) | GPU Peak Mem (GB) |
|---|---|---|
| zero-shot baseline | 3.05 | 5.8 |
| pre-filtering (QE) | 6.22 | 43.9 |
| interpolation | 26.17 | 44.2 |
| grad mt | 19.80 | 6.4 |
| path sampling | 7.00 | 6.0 |
| post-filtering (QE) | 6.22 | 49.9 |

Table 6: The computational cost (execution time and memory) for 50-shot/15-shot/cumulative 7-shot baseline. Cumulative 7-shot refers to translating while providing 0-shot, 1-shot to 7-shot examples, sequentially. The statistics are calculated running 10 randomly sampled sentences from FLORES test set into Hindi using Llama-3.1-8b.

| Few-shot # | Time per sentence (s) | GPU Peak Mem (GB) |
|---|---|---|
| 50 | 4.36 | 9.7 |
| 15 | 3.31 | 6.99 |
| 7-Cumulative | 23.80 | 6.4 |
| 2-Cumulative | 8.76 | 6.0 |

Table 7: Estimated time for possible filtering strategies of each method. N: # of sentences, M: # of sentences after pre-filtering, M=N*0.45 according to statistics from experiments.

| Method | Scenario | Estimation |
|---|---|---|
| IntGrad | **All**: Interpolation $\rightarrow$ Grad MT | $(26.17+19.80) \cdot N = \mathbf{45.97}N$ |
| | **Post-Filtering**: Baseline(0shot) $\rightarrow$ QE on baseline results $\rightarrow$ Interpolation $\rightarrow$ Grad MT $\rightarrow$ QE on GradMT results | $(3.05+6.22+26.17+19.80+6.22) \cdot N = \mathbf{61.46}N$ |
| | **Pre-Filtering**: Baseline $\rightarrow$ QE on baseline results $\rightarrow$ Interpolation $\rightarrow$ Grad MT | $(3.05+6.22) \cdot N + (26.17 + 19.80) \cdot M = \mathbf{29.96}N$ |
| | **Pre & Post-Filtering**: Baseline $\rightarrow$ QE on baseline results $\rightarrow$ Interpolation $\rightarrow$ Grad MT $\rightarrow$ QE on GradMT results | $(3.05+6.22) \cdot N + (26.17 + 19.80 + 6.22) \cdot M = \mathbf{32.76}N$ |
| Path Sampling | **All** | $(26.17+7.00) \cdot N = \mathbf{33.17}N$ |
| | **Post-Filtering** | $(3.05+6.22+26.17+7.00+6.22) \cdot N = \mathbf{48.66}N$ |
| | **Pre-Filtering** | $(3.05+6.22) \cdot N + (26.17 + 7.00) \cdot M = \mathbf{24.20}N$ |
| | **Pre & Post-Filtering** | $(3.05+6.22) \cdot N + (26.17 + 7.00 + 6.22) \cdot M = \mathbf{27.00}N$ |
| 50-Shot Baseline | | $\mathbf{4.36}N$ |
| 15-Shot Baseline | | $\mathbf{3.31}N$ |
| 7-Cumulative-Shot Baseline | | $\mathbf{23.80}N$ |
| 2-Cumulative-Shot Baseline | | $\mathbf{8.76}N$ |

# D  PROMPTS AND SETTINGS FOR TRANSLATION

**ChatGPT**  ChatGPT(gpt-3.5-turbo-0125) was used via API for translation with the same prompt from OpenAI's official documentation.[5] Temperature and top_p were set to 0.3 and 1, respectively. The actual prompt is as follows:

> **System:** You will be provided with a sentence in English, and your task is to translate it into ⟨ Target Language ⟩.
>
> **User:** ⟨ Sentence ⟩

**Llama-3.1 70B & 8B**  Llama-3.1 Instruct models were run on one A6000 GPU, using transformers library. 70B model were 4-bit quantized.Temperature and top_p were set to 0.6 and 0.9, respectively. The actual prompt is as follows:

> **System:** You will be provided with a sentence in English, and your task is to translate it into ⟨ Target Language ⟩.
>
> **User:** ⟨ Sentence ⟩

**Mistral-Nemo-Instruct-2407**  Mistral-Nemo-Instruct-2407 was run on one A6000 GPU, using transformers library. Temperature and top_p were set to 0.6 and 0.9, respectively. The actual prompt is as follows:

> **User:** You will be provided with a sentence in English, and your task is to translate it into ⟨ Target Language ⟩.
> Sentence: ⟨ Sentence ⟩

# E  SETTINGS FOR SENTENCE INTERPOLATION

For sentence interpolation Qwen2-72B-Instruct model was used. It was run on one A6000 GPU with 4-bit quantization using transformers library. Temperature and top_p were set to 0.6 and 0.9, respectively.

---

[5]https://platform.openai.com/docs/examples

# F  TEST RESULTS WITH DIFFERENT METRICS

We utilized MetricX (Juraska et al., 2023) and CometKiwi (Rei et al., 2023) to obtain supplementary metrics for evaluation. MetricX and CometKiwi are fine-tuned to predict MQM and DA scores, respectively. Tables 9 and 8 show the results from each model.

Table 8: CometKiwi results of IntGrad MT with the best ablation settings for different target languages.

| MT Model | Method | High Resource | | Mid Resource | | Low Resource | | |
|---|---|---|---|---|---|---|---|---|
| | | DE | ZH | HI | KO | SW | BN | MR |
| GPT 3.5 | 0 shot | 86.12 | 85.84 | 69.12 | 87.73 | 83.24 | 67.93 | 56.23 |
| | 15 shot | 86.71 | 86.71 | 69.72 | 88.85 | 82.99 | 69.65 | 58.37 |
| | 50 shot | 86.61 | 86.81 | 69.30 | 89.18 | 83.42 | 67.98 | 57.32 |
| | Intgrad MT$_{Post}$ | **87.33** | **87.79** | **73.15** | **90.49** | **85.41** | **73.66** | **63.52** |
| | Intgrad MT$_{Pre\&Post}$ | 87.07 | **87.79** | 73.12 | 90.26 | 85.08 | 72.72 | 60.65 |
| Llama 70B | 0 shot | 84.97 | - | 74.55 | - | - | - | - |
| | 15 shot | 84.11 | - | 72.94 | - | - | - | - |
| | 50 shot | 82.70 | - | 68.28 | - | - | - | - |
| | Intgrad MT$_{Post}$ | **86.90** | - | **77.13** | - | - | - | - |
| | Intgrad MT$_{Pre\&Post}$ | 85.61 | - | 75.95 | - | - | - | - |
| Llama 8B | 0 shot | 80.06 | - | 66.71 | - | - | - | - |
| | 15 shot | 81.64 | - | 69.58 | - | - | - | - |
| | 50 shot | 81.95 | - | 69.05 | - | - | - | - |
| | Intgrad MT$_{Post}$ | **84.71** | - | **73.64** | - | - | - | - |
| | Intgrad MT$_{Pre\&Post}$ | 80.63 | - | 73.24 | - | - | - | - |
| Mistral Nemo | 0 shot | 83.32 | 82.90 | 59.72 | 78.41 | 42.72 | 62.03 | 52.53 |
| | 15 shot | 85.46 | 85.73 | 66.77 | 87.48 | 47.51 | 72.81 | 63.73 |
| | 50 shot | 85.30 | 86.04 | 66.78 | 87.08 | 47.61 | 72.83 | 63.09 |
| | Intgrad MT$_{Post}$ | **87.09** | **87.13** | **70.61** | **89.35** | **54.46** | **77.91** | **67.89** |
| | Intgrad MT$_{Pre\&Post}$ | 87.00 | **87.13** | 70.22 | 89.32 | 53.39 | 77.76 | 67.77 |
| NLLB | - | 81.29 | 55.59 | 74.90 | 86.11 | 78.26 | 78.77 | 72.51 |
| TowerInstruct | - | 85.05 | 85.04 | - | 89.40 | - | - | - |

Table 9: MetricX results of IntGrad MT with the best ablation settings for different target languages.

| MT Model | Method | High Resource | | Mid Resource | | Low Resource | | |
|---|---|---|---|---|---|---|---|---|
| | | DE | ZH | HI | KO | SW | BN | MR |
| GPT 3.5 | 0 shot | 0.59 | 1.10 | 1.22 | 0.60 | 1.32 | 2.28 | 2.49 |
| | 15 shot | **0.54** | 1.04 | 1.18 | 0.52 | 1.31 | 2.10 | 2.16 |
| | 50 shot | 0.55 | 1.05 | 1.13 | 0.50 | 1.30 | 2.22 | 2.36 |
| | Intgrad MT$_{Post}$ | 0.54 | **1.03** | **0.91** | **0.44** | **1.18** | **1.55** | **1.42** |
| | Intgrad MT$_{Pre\&Post}$ | 0.55 | 1.09 | 0.92 | **0.44** | 1.18 | 1.59 | 1.45 |
| Llama 70B | 0 shot | 0.66 | - | 0.91 | - | - | - | - |
| | 15 shot | 0.71 | - | 0.97 | - | - | - | - |
| | 50 shot | 0.79 | - | 1.26 | - | - | - | - |
| | Intgrad MT$_{Post}$ | **0.58** | - | **0.68** | - | - | - | - |
| | Intgrad MT$_{Pre\&Post}$ | 0.66 | - | 0.68 | - | - | - | - |
| Llama 8B | 0 shot | 1.09 | - | 1.33 | - | - | - | - |
| | 15 shot | **0.80** | - | 1.10 | - | - | - | - |
| | 50 shot | 0.72 | - | 1.09 | - | - | - | - |
| | Intgrad MT$_{Post}$ | 0.95 | - | **0.90** | - | - | - | - |
| | Intgrad MT$_{Pre\&Post}$ | 1.02 | - | 0.92 | - | - | - | - |
| Mistral Nemo | 0 shot | 0.69 | 1.28 | 1.66 | 1.21 | 7.21 | 1.91 | 2.63 |
| | 15 shot | 0.58 | **1.07** | 1.24 | 0.55 | 6.40 | 1.64 | 1.68 |
| | 50 shot | 0.58 | 1.07 | 1.27 | 0.56 | 6.55 | 1.69 | 1.88 |
| | Intgrad MT$_{Post}$ | **0.54** | 1.09 | **1.11** | **0.54** | **5.45** | **1.13** | **1.26** |
| | Intgrad MT$_{Pre\&Post}$ | 0.55 | 1.09 | 1.13 | 0.54 | 5.56 | 1.13 | 1.26 |
| NLLB | - | 1.66 | 8.25 | 1.05 | 0.79 | 2.10 | 1.39 | 1.82 |
| TowerInstruct | - | 0.62 | 1.11 | - | 0.47 | - | - | - |

Table 10: BLEURT results of IntGrad MT with the best ablation settings for different target languages.

| MT Model | Method | High Resource | | Mid Resource | | Low Resource | | |
|---|---|---|---|---|---|---|---|---|
| | | DE | ZH | HI | KO | SW | BN | MR |
| GPT 3.5 | 0 shot | 78.58 | 73.70 | 68.31 | 68.77 | 75.80 | 67.95 | 68.03 |
| | 15 shot | **79.27** | 74.17 | 68.49 | 69.41 | 75.84 | 68.47 | 69.29 |
| | 50 shot | 79.11 | 74.20 | 68.74 | 69.54 | 76.13 | 67.82 | 68.57 |
| | Intgrad MT$_{Post}$ | 79.13 | **74.23** | **70.06** | **70.37** | **76.48** | **71.07** | **71.14** |
| | Intgrad MT$_{Pre\&Post}$ | 78.98 | 73.83 | 69.98 | 70.14 | 76.28 | 70.72 | 70.98 |
| Llama 70B | 0 shot | 77.57 | - | 71.34 | - | - | - | - |
| | 15 shot | 77.12 | - | 70.60 | - | - | - | - |
| | 50 shot | 76.10 | - | 68.68 | - | - | - | - |
| | Intgrad MT$_{Post}$ | **78.71** | - | 72.17 | - | - | - | - |
| | Intgrad MT$_{Pre\&Post}$ | 77.63 | - | **72.19** | - | - | - | - |
| Llama 8B | 0 shot | 74.69 | - | 66.96 | - | - | - | - |
| | 15 shot | 75.69 | - | 68.54 | - | - | - | - |
| | 50 shot | 75.79 | - | 68.41 | - | - | - | - |
| | Intgrad MT$_{Post}$ | **76.94** | - | **68.90** | - | - | - | - |
| | Intgrad MT$_{Pre\&Post}$ | 74.94 | - | 68.75 | - | - | - | - |
| Mistral Nemo | 0 shot | 76.27 | 69.87 | 60.95 | 59.13 | 51.51 | 62.29 | 64.20 |
| | 15 shot | 77.95 | 72.05 | 66.52 | 66.94 | 54.72 | 70.14 | 70.75 |
| | 50 shot | 77.89 | **72.41** | 66.47 | 66.49 | 55.19 | 70.15 | 70.46 |
| | Intgrad MT$_{Post}$ | **78.18** | 71.38 | **66.85** | 67.11 | **56.34** | **71.81** | **71.44** |
| | Intgrad MT$_{Pre\&Post}$ | 78.12 | 71.38 | 66.61 | **67.19** | 55.74 | 71.74 | 71.33 |
| NLLB | - | 76.88 | 58.09 | 72.17 | 67.40 | 73.84 | 75.92 | 76.19 |
| TowerInstruct | - | 78.04 | 72.53 | - | 69.96 | - | - | - |

Table 11: BLEU results of IntGrad MT with the best ablation settings for different target languages.

| MT Model | Method | High Resource | | Mid Resource | | Low Resource | | |
|---|---|---|---|---|---|---|---|---|
| | | DE | ZH | HI | KO | SW | BN | MR |
| GPT 3.5 | 0 shot | 40.45 | 45.58 | 23.06 | 27.86 | 32.93 | 9.99 | 5.94 |
| | 15 shot | 40.86 | 46.02 | 22.98 | 28.81 | 33.80 | 9.67 | 7.54 |
| | 50 shot | 40.96 | 45.98 | 23.52 | 28.90 | **33.95** | 9.75 | 6.08 |
| | Intgrad MT$_{Post}$ | **40.97** | **46.21** | **23.86** | **29.16** | 33.62 | **10.88** | 7.92 |
| | Intgrad MT$_{Pre\&Post}$ | 40.72 | 45.65 | 23.79 | 28.80 | 33.45 | 10.73 | **7.96** |
| Llama 70B | 0 shot | 38.71 | - | **29.09** | - | - | - | - |
| | 15 shot | 37.33 | - | 25.54 | - | - | - | - |
| | 50 shot | 35.39 | - | 22.84 | - | - | - | - |
| | Intgrad MT$_{Post}$ | 38.01 | - | 27.18 | - | - | - | - |
| | Intgrad MT$_{Pre\&Post}$ | **38.73** | - | 27.34 | - | - | - | - |
| Llama 8B | 0 shot | 30.81 | - | 21.54 | - | - | - | - |
| | 15 shot | 32.13 | - | 22.14 | - | - | - | - |
| | 50 shot | **32.61** | - | **22.37** | - | - | - | - |
| | Intgrad MT$_{Post}$ | 29.07 | - | 21.19 | - | - | - | - |
| | Intgrad MT$_{Pre\&Post}$ | 30.98 | - | 21.53 | - | - | - | - |
| Mistral Nemo | 0 shot | 35.70 | 38.83 | 17.47 | 20.17 | 12.13 | 8.35 | 5.31 |
| | 15 shot | **36.89** | 40.65 | 20.88 | 25.08 | **14.40** | 11.03 | 7.30 |
| | 50 shot | 36.72 | **41.15** | 21.05 | 25.28 | 13.75 | 11.16 | 7.69 |
| | Intgrad MT$_{Post}$ | 35.52 | 36.91 | 18.92 | 24.06 | 13.50 | 10.31 | 6.94 |
| | Intgrad MT$_{Pre\&Post}$ | 35.46 | 36.91 | 18.97 | 24.28 | 13.37 | 10.17 | 6.91 |
| NLLB | - | 37.56 | 26.96 | 32.71 | 28.44 | 31.49 | 16.82 | 15.70 |
| TowerInstruct | - | 39.01 | 41.80 | - | 30.15 | - | - | - |

# G  RESULTS FROM ABLATION

Table 12 shows the results for every combination of strategies that we explored in the ablation study(§6.1) with En-Ko translation. Table 13 presents the average results and changes in xCOMET scores for each combination of start selection strategies, the number of start sentences, and MT aggregation strategies.

Table 12: Full results on ablation study with En-Ko translation task. All scores are measured with DA score by xCOMET. 'Sort', 'Filter', and 'Tops' denote the start sentence selection methods. Each letter in parentheses denotes a similarity metric, with the order indicating the priority of the metrics. 'L' stands for Levenshtein distance, 'T' stands for tree edit distance, and 'S' stands for SBERT similarity. The highest results for each start selection strategy are highlighted in bold, while the second-highest results are underlined.

| Start Selection | Filtering | Aggregation | | | Baseline (3shot) |
| --- | --- | --- | --- | --- | --- |
| | | None | Poll | Prompt | |
| Sort (S-T) | All | 90.69 | 90.42 | 91.22 | 90.26 |
| | Pre | 91.31 | 91.14 | 91.50 | |
| | Post | 92.29 | 92.18 | **92.54** | |
| | Pre&Post | 91.70 | 91.84 | 92.08 | |
| Sort (T-S) | All | 89.93 | 90.02 | 90.93 | 90.57 |
| | Pre | 90.31 | 90.71 | 91.24 | |
| | Post | 91.81 | 91.97 | **92.47** | |
| | Pre&Post | 91.74 | 91.64 | 91.98 | |
| Sort (L-S) | All | 90.38 | 90.27 | 90.86 | 90.27 |
| | Pre | 91.10 | 90.95 | 91.12 | |
| | Post | 92.08 | 91.96 | **92.41** | |
| | Pre&Post | 91.83 | 91.68 | 91.91 | |
| Sort (T-L-S) | All | 90.17 | 89.99 | 91.32 | 90.44 |
| | Pre | 90.54 | 90.29 | 91.53 | |
| | Post | 92.06 | 91.90 | **92.69** | |
| | Pre&Post | 92.00 | 91.81 | 92.20 | |
| Sort (L-T-S) | All | 90.13 | 90.11 | 90.77 | 90.38 |
| | Pre | 90.69 | 90.85 | 91.13 | |
| | Post | 91.99 | 92.05 | **92.24** | |
| | Pre&Post | 91.93 | 91.82 | 91.86 | |
| SBERT Filter + Sort (T-L) | All | 90.36 | 90.23 | 91.07 | 90.35 |
| | Pre | 91.04 | 90.88 | 91.30 | |
| | Post | 92.07 | 92.10 | **92.53** | |
| | Pre&Post | 91.70 | 91.69 | 92.00 | |
| SBERT Filter + Sort (L-T) | All | 90.92 | 90.55 | 91.19 | 90.06 |
| | Pre | 91.46 | 90.98 | 91.23 | |
| | Post | 92.37 | 92.29 | **92.61** | |
| | Pre&Post | 91.95 | 91.12 | 91.34 | |
| Tops | All | - | 90.12 | 90.98 | 90.10 |
| | Pre | - | 90.65 | 91.30 | |
| | Post | - | 92.10 | **92.46** | |
| | Pre&Post | - | 92.06 | 92.01 | |

Table 13: Average xCOMET scores and score changes of selected outputs when applying every combination of strategies for start selection and aggregation in En-Ko translation. 'Score change' is calculated only for the adopted outputs. 'All' selects every output. 'Pre-filtering', denoted as 'Pre', applies zero-shot translation first and uses IntGrad MT only when the CometKiwi score is below a threshold. 'Post-filtering', denoted ans 'Post', selects outputs only if they outperform zero-shot translations. 'Pre- & Post- filtering', denoted as 'Pre & Post' combines these two strategies. We also report the number of end sentences for which interpolation and Gradual MT is executed ('No. of Interpolated End Sents') and the number of end sentences for which the IntGrad MT output is selected over zero-shot translation ('No. of Selected Outputs'). The results show that 'Pre & Post' reduces the number of interpolation by more than half while maintaining nearly the same translation performance.

| Output Adoption Strategy | Avg. Score | Avg. Score Change of Selected Outputs | No. of Interpolated Sents (%) | No. of Selected Outputs (%) |
|---|---|---|---|---|
| Zeroshot | 89.48 | - | - | - |
| All | 90.55 | 1.06 | 21864 (100%) | 21864 (100%) |
| Pre | 91.01 | 3.70 | 10468 (48%) | 10468 (48%) |
| Post | **92.21** | 5.30 | 21864 (100%) | 11270 (52%) |
| Pre&Post | 91.81 | **7.98** | 10468 (48%) | 6664 (30%) |

# H    Result from Experiments to Save Computational Cost of Gradual MT

Table 14: BLEU scores of IntGrad MT with different strategies to save computational cost of gradual MT. 'Default' denotes the strategy which uses every sentences from interpolation path. 'Path truncation' denotes the strategy which uses three recent translations for gradual MT. 'Path sampling' denotes the strategy which uses start, middle and end sentence from the interpolation path for gradual MT.

| SETTING | DE | ZH | HI | KO | SW | BN | MR |
|---|---|---|---|---|---|---|---|
| **15 Shot** | 40.86 | 46.02 | 22.98 | **28.81** | **33.80** | 9.67 | 7.54 |
| **Default** | 40.84 | 46.07 | 23.24 | 28.69 | 33.61 | 9.10 | 6.54 |
| +Pre | 40.72 | 46.07 | 23.25 | 28.39 | 33.15 | 9.30 | 7.20 |
| +Post | 40.77 | **46.37** | **23.87** | 28.70 | 33.09 | **10.81** | 7.81 |
| +Pre & Post | 40.79 | 46.37 | 23.84 | 28.35 | 33.01 | 10.57 | 7.63 |
| **Path Truncation** | 40.46 | 45.01 | 22.53 | 28.33 | 33.06 | 10.57 | 6.97 |
| +Pre | 40.45 | 45.58 | 23.06 | 27.87 | 32.93 | 9.99 | 5.94 |
| +Post | **40.87** | 46.06 | 23.48 | 28.78 | 32.86 | 10.67 | 8.01 |
| +Pre & Post | 40.45 | 45.58 | 23.06 | 27.87 | 32.93 | 9.99 | 5.94 |
| **Path Sampling** | 40.39 | 45.51 | 22.85 | 28.27 | 33.11 | 9.73 | 7.17 |
| +Pre | 40.57 | 45.76 | 23.06 | 28.04 | 32.99 | 9.67 | 7.51 |
| +Post | 40.78 | 45.98 | 23.69 | 28.71 | 33.07 | 10.35 | **8.14** |
| +Pre & Post | 40.69 | 45.84 | 23.62 | 28.04 | 32.95 | 10.21 | 7.95 |

# I    Samples of interpolation paths

We sampled 100 interpolation paths for En-Ko translation containing more than three sentences and obtained sentence embeddings using SBERT. Each interpolation path was then plotted on a 2D plane using Principal Component Analysis (PCA). We identified four notable patterns, which we named based on their shapes: Arc, Triangle, Zig-Zag, and Leap. The Arc pattern was the most common. In this pattern, the sentences gradually shift toward the end sentence, following an arc-shaped trajectory. While the Arc pattern presents a relatively smooth shape, the Zig-Zag and Triangle patterns exhibit more spiky trajectories, though the sentences still progress toward the end sentence. The final pattern, Leap, shows a single large shift toward the end sentence, without the gradual progression seen in the other patterns. Although the Leap pattern might initially be perceived as a "bad interpolation," closer examination reveals that similar intervals also appear in other patterns. Overall, our qualitative analysis showed that, regardless of the pattern, sentence interpolation typically shifts the start sentence toward the end sentence. Figure 6 shows 24 samples from these 100 paths.

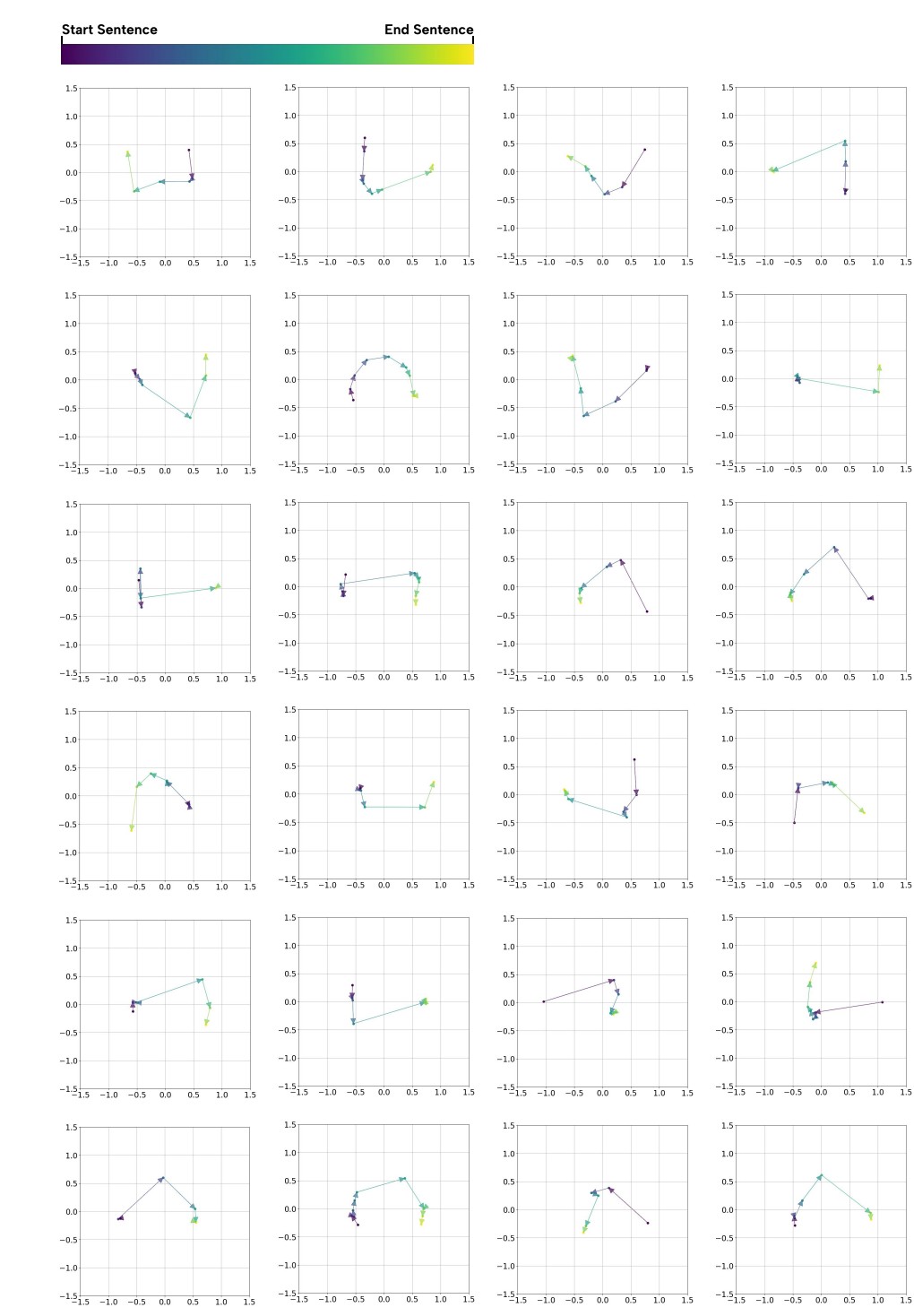

Figure 6: 24 samples of 2D scatter plots. The plots are projected from SBERT embeddings onto a 2D plane using PCA. The X and Y axes of each plot represent the first and second principal components, respectively. Arrows in each plot show the trajectory of sentence shifts from the start sentence (blue-colored dot) to the end sentence (yellow-colored dot).

