# OpenReview forum: "IntGrad MT: Enhancing LLMs' Machine Translation Capabilities with Sentence Interpolation Guided Gradual MT"
_ICLR.cc/2025/Conference — Submitted to ICLR 2025_

### Official Review · Reviewer_97xN · 2024-10-29

**Soundness:** 3
**Presentation:** 3
**Contribution:** 4
**Rating:** 6
**Confidence:** 4

**Summary:**

This paper addresses the topic of improving LLM-based machine translation through prompting. Specifically, the paper introduces IntGrad MT, which is a combination of sentence interpolation and gradual machine translation. The intuition is to start from translating a sentence that the model can translate well, and gradually interpolate that sentence into the desired source sentence using an LLM. The LLM used for translation, then, is prompted to translate each of the interpolated sentences, given its previous translations as a guide.

Update after discussion period: The revised version of the paper has partially addressed my concerns about baselines and evaluation metrics, so I am raising my score.

**Strengths:**

The proposed IntGrad method is a very interesting and novel idea. It is unlike any prior work that I am familiar with, and (with some improvements to the experiments) I could see opening up new avenues for research into LLM prompting.

**Weaknesses:**

1. **Evaluation metrics**. The proposed model is evaluated using mostly xCOMET. I have two potential concerns here: a) Although xCOMET correlates well with human judgments on *high-resource* languages, it is much less reliable on *low-resource* languages. In particular, the DA data used to train xCOMET only covers EN-ZH and EN-DE of the evaluated language pairs. b) The models partially develop on COMET, as it is used in output selection (CometKiwi but the base model and training data is largely the same) and sentence pool creation. This raises the concern that gains could be an artifact of tuning towards the metric, and not necessarily due to better translation. A simple fix could be to use a model-free metric (which could be more language-agnostic) as a point of comparison (e.g., chrF), and to include even a small amount of human evaluation to corroborate the results.

2. **Weak baselines**. The paper (L16) points out that several methods have been proposed for improving LLM low-resource translation including few-shot translation and using dictionaries and grammar books, but does not do any comparison to using dictionaries and grammars. In addition, for low-resource translation neural machine translation is a strong baseline and should be included, but no rationale is given for why the paper doesn't compare to NMT or why LLM-based translation would be preferable (it's not cheaper or faster, either). The zero-shot and three-shot baselines are not particularly fair either: given how compute-intensive the proposed method is, it would be fair to use more shots (e.g., 5 or 10) or chain-of-thought. In addition, per L265, "we use the same start sentences for interpolation as in the few-shot examples", which also seems unfair to few-shot, as the extra compute and iterations used in the proposed IntGrad could be similarly used to find good/appropriate few-shot examples.

3. **Construction of the sentence pool**. The construction of the sentence pool seems it would have a huge effect on the final performance, and it seems underexplored in this paper. In particular, I am curious what the impact of the size and quality of the sentence pool is. Also, the sentence pool used in the paper is the development set corresponding to the test sets, so it is quite well-matched to the test set in domain, style, and format. This is not a realistic setup, and it would be good to see experiments showing what happens when you don't know the domain in advance and have to use a generic sentence pool, as in a real-world application.

4. **Practicality of the method**. I would have liked to see some discussion of translation time and cost, beyond a cursory mention in limitations (L534 "introduces some computational overhead"). What computational overhead, what is the increase in cost, and how much longer do translations take? Is it possible to batch the translations? These are relevant questions to be explored in at least some depth, given that I think it's quite a large increase.

**Questions:**

1. Although you mention in limitations that ﻿﻿﻿"sentence interpolation did not perform well in languages other than English" (L537), I'd be curious to see examples of others, and some evaluation results (even presented as negative results). I think that would be informative for future work.

2. L252: "We first utilize the dev split of the dataset to create start sentence pool. During evaluation, to test the effect of the output selection strategy, we selected 10% of the test portion of the dataset to set the DA score threshold. The remaining 90% is used to assess the overall performance on the dataset": since these are established dev/test splits, I recommend using a 10% subset of the *dev* set to set the DA score threshold. Maintaining the test split as it was originally released and evaluating on the whole thing will help reproducibility.

3. Do you think you could eventually interpolate from one language to another? E.g., if a model translates English->Hindi well but German->Hindi poorly, do you think you could gradually interpolate English source with German source to eventually get a German sentence? This is not essential for the current paper, of course, but I'm curious to hear your intuition on this.

---

> ### Author Response · Authors · 2024-11-27
>
> Dear reviewer, thank you for providing great ideas to improve our study!
>
>
> ### About Baselines and Metrics
> Please refer to the general comments section for a detailed discussion on this topic.
>
>
> ### About Using the Dev Set as a Source for the Start Sentence Pool
> We understand your concern about using sentences from the development set, as they are well-aligned with the test set. However, upon examining the interpolation paths, we observed that the start and end sentences often show significant differences in terms of topics or styles. This suggests that domain alignment is not the sole factor influencing performance improvement. Additionally, please note that our baseline few-shot translations also utilized the dev set, yet they did not achieve the same performance improvement as ours.
>
> ### Interpolation in Other Languages
> Your question regarding interpolation in non-English languages is relevant. Unfortunately, we have not conducted separate evaluations on interpolation applied to non-English sentences. Our initial tests using the ChatGPT API revealed limitations at the time. For example, ChatGPT often responded that it was "impossible to transform one sentence into another" in non-English languages or generated nonsensical sentences.
> Although we did not pursue further evaluation at that stage, newer and larger models seem to demonstrate some interpolation capabilities in non-English languages. For example, we have observed promising results with GPT-4O. We provide examples of interpolation paths generated by GPT-4O, using the same start and end sentences as in Appendix A.3. in the comment below.
>
> ### About Splitting the Test Set
> Your point about reproducibility is well-taken. We are happy to incorporate this change for the camera-ready version. However, due to the numerous experiments we are conducting during the discussion phase, we regret that we are unable to provide the requested result at this stage. We would also like to emphasize that while we agree that using the same test set would facilitate comparisons with future work, we have ensured fairness by comparing the baselines on an identical test set split.
>
>
> ### About Interpolating to Another Language
> Your question about interpolation between different languages is fascinating. We believe that interpolation and the subsequent GradMT process could indeed be extended to translate from an easier language pair to a more challenging one.
> Our assumption is that the relationships between the source sentences (i.e., the sentences along the interpolation path) provide critical hints for translating the end sentence. Therefore, the success of IntGrad MT is influenced by the quality of the language representations within the LLM. If two languages share strong linguistic similarities (e.g., syntactic or semantic features) and the LLM has been trained on a substantial number of parallel pairs for that language pair, interpolation and IntGrad MT are more likely to perform effectively. We would be happy to apply this idea in our future work.
>
>
>
>
> ### About the Effects of Start Pool Quality
> Your question about the impact of start pool quality is insightful. In our experiments, we arbitrarily set the size of the start sentence pool to 100, and the majority of the quality estimation (QE) scores for the start sentences were 1.0. Consequently, we did not extensively investigate the effect of start sentence quality.
> However, one of our subsequent experiments, not included in the paper, partially addresses this question. In this experiment, we employed a few-shot approach to construct the start sentence pool. Specifically, we provided the model with three similar sentences retrieved from the FLORES-200 development dataset, along with their gold translations, as few-shot examples. This approach likely improved the quality of translations within the start sentence pool.
> The evaluation results from this experiment showed marginal improvements for low-resource languages, suggesting that the quality of the start sentence pool may indeed influence the performance of IntGrad MT. Please refer to the table in the next comment.

---

> > ### Author Response · Authors · 2024-11-27
> >
> > **Table: xCOMET scores of IntGrad MT with different strategies for building the start pool. Experiments were conducted using gpt-3.5-turbo-0125.**
> > | SETTING                                     | DE   | ZH   | HI   | KO   | SW   | BN   | MR   |
> > |---------------------------------------------|------|------|------|------|------|------|------|
> > | BASELINE 15 Shot                            | 0.98 | 0.92 | 0.73 | 0.91 | 0.82 | 0.70 | 0.46 |
> > | BASELINE 50 Shot                            | 0.98 | 0.92 | 0.73 | 0.91 | 0.82 | 0.68 | 0.45 |
> > |---------------------------------------------|------|------|------|------|------|------|------|
> > | 1 Start (Previous)                          | 0.98 | 0.91 | 0.74 | 0.91 | 0.82 | 0.70 | 0.47 |
> > | +Thr.                                       | 0.98 | 0.92 | 0.75 | 0.91 | 0.82 | 0.72 | 0.49 |
> > | +Delta                                      | 0.98 | 0.93 | 0.76 | 0.92 | 0.84 | 0.74 | 0.52 |
> > | +Thr. + Delta                               | 0.98 | 0.92 | 0.76 | 0.92 | 0.84 | 0.74 | 0.51 |
> > |---------------------------------------------|------|------|------|------|------|------|------|
> > | 1 Start (w/ Few-shot implemented Start pool)| 0.98 | 0.92 | 0.74 | 0.91 | 0.82 | 0.70 | 0.48 |
> > | +Thr.                                       | 0.98 | 0.92 | 0.75 | 0.91 | 0.82 | 0.72 | 0.49 |
> > | +Delta                                      | 0.98 | 0.93 | 0.77 | 0.92 | 0.84 | 0.75 | 0.52 |
> > | +Thr. + Delta                               | 0.98 | 0.92 | 0.77 | 0.92 | 0.84 | 0.74 | 0.50 |
> > |---------------------------------------------|------|------|------|------|------|------|------|
> >
> > **Interpolation Paths in other languages**
> >
> > Korean
> > - '이 케이크는 자메이카에서 \u200b\u200b가장 인기 있는 디저트 중 하나가 되었으며, 다양한 매장에서 흔히 볼 수 있습니다.  '
> > - '어떤 보험 상품은 인기 있는 디저트의 제네릭 버전만 보장할 수 있지만, 특정 제품에 대한 보장은 여전히 가능합니다.  '
> > - '일부 상품은 인기 있는 디저트의 제네릭 버전만 제공할 수 있지만, 특정 옵션은 여전히 필요할 수 있습니다.  '
> > - '일부 보험 상품은 특정 제품의 제네릭 버전만 보장할 수 있지만, 환자는 여전히 특정 옵션을 받을 권리가 있습니다.  '
> > - '일부 보험 상품은 특정 피임약의 제네릭 버전만 보장할 수도 있지만, 환자는 여전히 의료 제공자가 의학적으로 필요하다고 판단하는 특정 제품에 대한 보장을 받을 권리가 있습니다.  '
> >
> > Swahili
> > - 'Keki hiyo imekuwa mojawapo ya desserts maarufu zaidi ya Jamaika, mara nyingi hupatikana katika maduka mbalimbali.  '
> > - 'Baadhi ya desserts maarufu zaidi ya Jamaika yanaweza kupatikana katika maduka, lakini kuna vionjo vya kipekee ambavyo vinajumuishwa mara nyingine.  '
> > - 'Baadhi ya desserts maarufu yanaweza kupatikana kwa matoleo ya kawaida tu katika maduka, lakini kuna aina maalum ambazo watu wanazihitaji.  '
> > - 'Baadhi ya mipango ya desserts maarufu inaweza kujumuisha matoleo ya kawaida tu, lakini kuna vitu maalum ambavyo watu wanajua ni muhimu.  '
> > - 'Baadhi ya mipango inaweza kujumuisha matoleo ya kawaida tu ya baadhi ya vitu maarufu, lakini watu bado wana haki ya kupata matoleo maalum ambayo yanatambulika kama muhimu.  '
> > - 'Baadhi ya mipango inaweza kujumuisha matoleo ya kawaida tu ya baadhi ya vidhibiti mimba, lakini wagonjwa bado wana haki ya kupata huduma mahususi ambayo watoa huduma wao wanaona ni muhimu kiafya.'
> >
> > Bengali
> > - 'কেকটি জ্যামাইকার অন্যতম জনপ্রিয় ডেজার্ট হয়ে উঠেছে, যা প্রায়শই বিভিন্ন দোকানে পাওয়া যায়।  '
> > - 'কিছু জনপ্রিয় ডেজার্ট, যেমন কেক, বিভিন্ন দোকানে পাওয়া যায়, কিন্তু কিছু ক্লাসিক সংস্করণ নির্দিষ্ট পরিকল্পনায় অন্তর্ভুক্ত না-ও হতে পারে।  '
> > - 'কিছু জনপ্রিয় ডেজার্টের প্রচলন রয়েছে, এবং মাঝে মাঝে কিছু নির্দিষ্ট সংস্করণগুলো সাধারণভাবে উপস্থিত থাকে।  '
> > - 'কিছু পরিকল্পনা কিছু নির্দিষ্ট জনপ্রিয় পণ্যের জেনেরিক সংস্করণ কভার করতে পারে, তবে অন্যান্য নির্দিষ্ট প্রকারও পাওয়া যায়।  '
> > - 'কিছু পরিকল্পনা কিছু নির্দিষ্ট পণ্যের জেনেরিক সংস্করণ কভার করতে পারে, তবে অন্যগুলি নির্দিষ্ট প্রয়োজনীয়তার ক্ষেত্রে অন্তর্ভুক্ত হয়েছে।  '
> > - 'কিছু পরিকল্পনা কিছু নির্দিষ্ট গর্ভনিরোধকের জেনেরিক সংস্করণগুলিকে কভার করতে পারে, তবে রোগীরা এখনও একটি নির্দিষ্ট পণ্যের কভারেজের অধিকারী যা তাদের প্রদানকারীরা চিকিৎসাগতভাবে প্রয়োজনীয় বলে মনে করেন।'

---

> ### Author Response · Authors · 2024-12-01
>
> Dear Reviewer 97xN,
>
> Thank you for expressing interest in our work and providing valuable feedback to enhance our paper. We also appreciate your recognition of the novelty of our approach, noting that it has not been attempted before.
>
> During the discussion period, we endeavored to address your concerns and strengthen our research.
>
> We have introduced additional baselines (15-shot, 50-shot, NLLB, TowerInstruct) and new metrics (BLEURT, BLEU), along with an analysis of computational costs. Furthermore, we explored methods to reduce the computational demands of Gradual MT—specifically through path sampling and path truncation. These efforts have successfully halved the computational costs and simultaneously improved performance.
>
> As the discussion period draws to a close, we hope you have had the opportunity to review our revised paper. Should you have any further suggestions that could improve our work, we would be grateful to hear your thoughts.

---

### Official Review · Reviewer_efMc · 2024-10-30

**Soundness:** 1
**Presentation:** 2
**Contribution:** 2
**Rating:** 3
**Confidence:** 5

**Summary:**

This paper proposes IntGrad MT, a method to enhance machine translation capabilities of large language models (LLMs) without additional training by using sentence interpolation and gradual MT. The approach constructs chains of sentences that incrementally increase in difficulty, using the model's own translations as few-shot examples. The authors evaluate their method on various LLMs (GPT-3.5, Mistral Nemo, Llama models) across multiple languages and report improvements especially for low-resource languages.

**Strengths:**

1. The paper addresses an important problem of improving LLM translation capabilities for low-resource languages *without* additional training

2. The evaluation includes multiple models. Even though source is fixed to English, the paper evaluates on multiple target languages from different language families and using different scripts.

3. The ablation study attempts to analyze different components of the method

**Weaknesses:**

1. A very big issue is that the baselines (0-shot, 3-shot) are unfairly chosen because they are too weak, and as a result I am not at all convinced that the method proposed by the authors would outperform stronger baselines. Since the main contribution of the paper is improving MT quality, this is something that definitely should be improved. At the absolute minimum, additional baseline results for n-shot should be included, where n is chosen such that the computational costs are similar to the method proposed by the paper. Ideally, some other popular methods that improve LLM-based MT without fine-tuning (from related work Section 2.1) will also be included.

Let's consider example 3 from the appendix, which has an interpolation path of 7 sentences. If I understand correctly, this means that we start from the first sentence and do a 0-shot translation, and then recursively do (n-1)-shot translations until we arrive at the final sentence in the interpolation path. The total costs for X are thus:

    - s1 -> 0-shot translation +
    - s2 -> 1-shot translation with input (s1) +
    - s3 -> 2-shot translation with input (s1,s2) +
    - s4 -> 3-shot translation with input (s1,s2,s3) +
    - s5 -> 4-shot translation with input (s1,s2,s3,s4) +
    - s6 -> 5-shot translation with input (s1,s2,s3,s5) +
    - s7 -> 6-shot translation with input (s1,s2,s3,s5,s6)

(Note that this ignores the costs for step 0: start sentence pool creation, step 1: start sentence selection, step 3: MT results aggregation. It also assumes a single start sentence, otherwise costs will be multiplied by number of start sentences (assuming same interpolation sentences).)

2. The method is very computationally expensive compared to the baselines. This is true even for much stronger baselines (e.g., n-shot with relatively large n) that are not included in the paper. The authors mention "IntGrad MT introduces some computational overhead", but this seems a severe understatement. The computational overhead of the method is not properly analyzed, but this analysis should have been included to help readers understand these trade-offs.

3. As acknowledged by the authors, their method only works when English is the source language. This is not necessarily a prohibitive issue, but when we have few-shot (with larger n) available as alternative, it is unclear why one would use this more complicated method over that (current results fail to convince me of the improvements over stronger baselines, see first point).

4. I would not call LLMs translation capability "inherent", as you do in the abstract. See e.g. https://aclanthology.org/2023.acl-long.524/, which shows that translation capabilities are mostly due to (incidental) bilingualism, i.e., parallel data.

5. While the authors briefly discuss related work, the discussion lacks depth. It is not clear how the authors method is different from this discussion. (It can be inferred by a reader who is knowledgeable about the related works, but that is insufficient.)

6. A lot of clarity and typo issues, I list some below:
    - Figure 1 is a bit unclear, nowhere is mentioned what blue/red colours mean. I think I can infer it but it is not clear enough.
    - line 89: "To what sentences IntGradMT can be effective?" awkward sentence, rephrase to "For which sentences is IntGradMT effective?"
    - line 135: "which is a prompting technique asks model" -> "that asks the model"?
    - line 214: "list of translation" -> "list of translations"
    - lots of issues with missing space before opening parenthesis, e.g. line 235: "Nemo(Mistral-Nemo"; line 281: "selection(step 1)"
    - line 372: "theee" -> "three"

**Questions:**

1. Baseline Comparison and Computational Cost
    - Given the computational complexity of your method (detailed below), why weren't stronger baselines included? At minimum, n-shot baselines with comparable computational cost should be included.
    - How do you justify this overhead compared to simpler approaches?
    - Why weren't comparisons made to other non-finetuning MT improvement methods mentioned in Section 2.1?

2. Limited Language Direction
    - The method only works with English as the source language. Given this limitation and the high computational cost, could you elaborate on why one should choose this method over simpler approaches like n-shot with larger n?
    - Have you explored why the method fails for non-English source languages?

3. Technical Claims and Methodology:
    - The paper describes LLMs' translation capability as "inherent", but research (e.g., https://aclanthology.org/2023.acl-long.524/) suggests it comes from parallel training data. Could you clarify this characterization?
    -How is your method fundamentally different from previous work? The related work section doesn't clearly distinguish your contributions.

4. Clarity Issues
    - Could you clarify the meaning of colors in Figure 1?
    - Several writing issues need addressing (e.g., "To what sentences IntGradMT can be effective?", missing spaces before parentheses, etc.)
    - Can you provide more precise technical details about the aggregation method and threshold determination?

5. Implementation Details:
    - How stable is the method across different interpolation models?
    - What is the variance in performance across multiple runs?
    - How were hyperparameters (like the number of start sentences) chosen?
    - It is likely that the reference-free QE model you're using does not work very well for low-resource languages. How do you inspect to what extent this is an issue?

---

> ### Author Response · Authors · 2024-11-27
>
> Dear reviewer, thank you for providing detailed feedback. Below, we address your comments point by point:
>
> ### About the Baselines and Computational Cost
> Please refer to the general comments for a detailed discussion on this topic.
>
>
>
>
> ### Comparison with Other Alternatives
> Our method is distinct from prior works in several key ways:
> - We identify which samples a translation model is good at translating and which samples it is bad at, and devise a framework to bring the knowledge of the former to the latter.
> - This framework uses the novel method interpolation, which creates a chain of sentences that link two already existing sentences. Pseudo-labels are generated for each link in this chain through GradMT, producing a chain of parallel sentence pairs varying in degree of translation difficulty.This produces more diverse data to put into the prompt than retrieving or generating few-shot examples based on similarity.
> - Our work does not require additional resources such as grammar books or dictionaries.
>
> To the best of our knowledge, the other alternatives using LLMs include providing few-shot examples and the 'Chain of Dictionary (CoD),' as presented in our related works. The results for CoD were significantly worse than ours, and the few-shot approach has its limitations — adding more examples does not enhance the LLM's capability. For instance, **GradMT with path sampling is equivalent to performing 2-shot and 3-shot translations sequentially, yet even 50-shot translations could not match our performance.** Furthermore, we outperform both TowerInstruct and NLLB, which are specifically trained for translation in mid-resource languages.
>
> This suggests that our approach is better than most existing baselines whether they utilize LLMs or not. Please refer to Table 1 for the results.
>
>
> ### About Limited Language Direction
> We presented results only for English-to-X translation because our interpolation model, Qwen, was unable to translate from other languages. However, experiments with a recent version of ChatGPT demonstrated its capability to interpolate from non-English languages. This suggests that our IntGrad MT framework can be extended to translation from other languages as well, which we leave as our future work. We believe the limitation of directionality is incurring from the limitation of the current interpolation capability (which improves day by day as we see with ChatGPT) rather than the limitation of our idea.
>
>
> ### Technical Claims and Methodology
> Thanks for providing a chance to clarify what we mean by “inherent”
> By “inherent translation capability,” we refer to the ability of LLMs to translate using the knowledge”, we meant LLM’s translation capability itself without external resources such as dictionaries or grammar guides. We do not intend to claim how LLM possesses its translation capabilities, but we are interested in maximizing LLM’s translation capability by referring its own translation results (thus inherent) by utilizing IntGradMT.
>
>
> ### Clarity Issues
> We appreciate your comments regarding clarity and will address these concerns in the revised version of the paper.
>
>
> ### Implementation Details
> In earlier experiments, we explored whether ChatGPT could serve as the interpolation model. The results were superior to Qwen’s performance; however, we did not integrate ChatGPT into the entire IntGrad pipeline due to practical consideration of cost.
>
> Regarding the QE model, we use Unbabel/wmt23-cometkiwi-da-xxl, which has demonstrated strong evaluation performance for many low-resource languages (see arXiv:2309.11925). We also provide results from various metrics, most of which show alignment with this QE model.
>
> The results presented in our paper are based on single-run statistics.

---

> ### Author Response · Authors · 2024-12-01
>
> Dear Reviewer efMc,
>
> Thank you once again for your detailed feedback and guidance on improving our paper. We also appreciate your recognition of our strengths, including our novel approach that enhances LLM translation capabilities without training, applicable across multiple models.
>
> We have strived to incorporate your feedback as thoroughly as possible, particularly concerning baselines and computational costs. We introduced stronger baselines (15-shot, 50-shot, NLLB, TowerInstruct) and demonstrated that merely increasing the number of few-shot instances does not sufficiently improve LLM translation capabilities. Additionally, we have provided a detailed analysis of computational costs to better clarify the trade-offs between cost and performance.
>
> Furthermore, we have included results from new experiments involving path sampling and path truncation, strategies aimed at reducing the costs of gradual MT. It is noteworthy that these new approaches have significantly enhanced performance while halving computational time.
>
> As the discussion period draws to a close, we kindly remind you to review our updated paper and share any further feedback to help us refine our work.
>
> Thank you.

---

### Official Review · Reviewer_Y21e · 2024-10-31

**Soundness:** 3
**Presentation:** 3
**Contribution:** 3
**Rating:** 8
**Confidence:** 5

**Summary:**

The paper introduces a method called IntGrad MT that alleviates performance of LLMs for MT in low-resource language pairs using parametric sources of information with no extra training. In IntGrad they utilise 2 techniques, sentence interpolation and gradual MT. Former starts from an easy seed sentence and keeps on changing it till it reaches the source sentence (sentence to translate) and latter keeps on translating each one while using the previous translation results as few-shot examples for the current sentence. The authors show efficacy of their method by conducting thorough experiments and ablation studies.

**Strengths:**

- The paper is very clearly written and easy to follow except for the Output selection part.
- Their approach doesn’t require extra training and utilises existing parametric information.
- The experiments performed by authors are quite extensive including ablation study for finding optimal combination of strategies and sentence interpolation analysis.
- The authors used xCOMET over COMET which has error span detection capabilities. I highly recommend them showing how their approach was better when compared to baselines by showing error spans of the baselines (which their own approach won’t have).
- Figure 1 gives a good intuition of IntGrad MT and Figure 2 is really helpful for visualising Gradual MT

**Weaknesses:**

- Output selection could have been worded properly. Although it is clear from the paragraph that “thr” is a pre-processing step and “delta” is a post-processing step but:
    - “thr” should come before sentence interpolation in Figure 3 which is not the case
    - “delta” should come before aggregation?
    - From what I understood - start sentences -> thr -> interpolation -> Gradual MT -> delta -> aggregation -> Final translation
    - If this is correct then you please modify Figure 3 and meaning of output adoption strategy
- Even though this approach doesn’t require training but interpolation step is quite costly as it uses a 72B LLM to generate sentences to reach end sentence which increases the latency of the overall translation. It is evident that "thr" reduces number of interpolated sentences but it increases the latency by how much compared to the baselines? How do you plan to reduce the latency of the Interpolation model?
- For few-shot baseline, authors use 3 MT pairs but for their thr + delta approach I can see from Table 3 that Interpolation and GradMT were executed for 45% sentences out of which number of selected outputs were 279 from which final MT result is produced. Do you ensure a fair comparison between you method and the 3-shot baseline, given the difference in the number of examples used.

**Questions:**

- I had some trouble understanding “Start Sentence Pool Creation”. Start sentences are essentially in English, then why were they translated as mentioned in Section 4.1?
- I can see the authors used Euclidian distance throughout the paper but cosine similarity is used more frequently when measuring similarity between 2 sentences using SBERT in many NLP papers. Why did the authors use Euclidian distance and not cosine similarity metric?
- Please change the colour of MQM(MetricX) for Figure 3 as it looks quite washed out when printed.
- I wanted to know if having bigger context length had any detrimental affect on the final results in Table 3?


Typos and formatting-
- Line 241: SBERT similarity
- Line 471: measure
- Table 1, 2, and 3 have a missing \toprule. Missing \bottomrule for Table 2
- Figure 3 says “n” start sentences but caption says “k” start sentences

---

> ### Author Response · Authors · 2024-11-27
>
> Dear Reviewer,
> Thank you very much for your insightful feedback and suggestions!
> Below, we address your feedbacks and questions in detail:
>
>
> ### Terminology
> To enhance clarity, we have replaced the terms “thr” with "pre-filtering" and “delta" with “post-filtering” to better reflect the stages at which these processes are applied. Figure 3 has been updated accordingly. Specifically:
> - Pre-filtering (“thr”) is applied between the creation of the start sentence pool and interpolation.
> - Post-filtering (“delta”) is applied after aggregation.
>
>
> ### Computation
> You are correct that interpolation is computationally intensive. To mitigate this, we employ quantization to reduce memory usage. In terms of latency, interpolation does represent a bottleneck in the overall pipeline of IntGrad MT. However, the pre-filtering process (formerly “thr”) significantly reduces the computational load. Although pre-filtering introduces additional latency due to running QE model (approximately 6 seconds per sentence) for every translation,  this enables already good translations to bypass the more expensive IntGradMT step. In our example, we bypass 55% of instances and the total estimated time for translating a sentence decreases, on average, from 45.97 to 29.96. For further details on computational efficiency, we kindly refer you to the general comments and Appendix C.
>
>
> ### Start Sentence Pool Creation
> The start sentences were translated using zero-shot translation and evaluated to ensure that the model performs reasonably well on this subset.
>
> ### On Using Euclidean Distance
> We chose Euclidean distance because it aligns with the core intuition of IntGrad MT, as illustrated in Figure 1 of the paper. Our method aims to construct a “path” that bridges the start and end sentences. Using Euclidean distance allowed us to verify this intuition indirectly.
>
>
> ### Color Adjustments in Figures
> Thank you for pointing this out! We will update the figure colors shortly to improve their clarity and visual appeal.
>
>
> ### Effect of Context Length
> We may have misunderstood your question regarding context length, as Table 3 focuses on the effect of filtering strategies (formerly “output selection strategies”) and is unrelated to context length. The table illustrates the impact of filtering on the number of source sentences processed by IntGrad MT.
> In general, longer context lengths may have a detrimental effect on the final results. Our recent experiments on path sampling show improved scores compared to the original IntGrad method. This may result from reduced noise due to shorter paths or the benefits of the shorter path itself. Differentiating between these two effects could be an interesting avenue for future research.

---

> > ### Comment · Reviewer_Y21e · 2024-11-27
> >
> > Thanks a lot for your explanations and also updating the pdf. Your approach looks quite clear now! Exploring Path Sampling and Path truncation is a good avenue for future work, however, trying to reduce the overall paths itself will benefit this approach the most in terms of latency of the interpolation step. Just last few bits:
> >
> > - Can you also show xCOMET scores of IntGrad MT with 7-Cumulative-Shot?
> > - Please mention latency part in another column? I think with Table 4 this will be a good to have. You can put latency calculations in Appendix though. Table 4 is more of an ablation (just re-iterating).
> > - Table 1, 2, 3. 4 have a missing \toprule
> >
> > I think authors have put in a lot of work for this paper and for the rebuttal too. Their approach sounds coherent and logical. I have updated my rating. Thank you and all the best!

---

> > > ### Author Response · Authors · 2024-11-30
> > >
> > > Dear reviewer, we greatly appreciate your updated rating and your comments regarding the importnace of strategies to reduce the latency of interpolation step. We will endeavor to idntify more efficient options for interpolation in our future work.
> > > Below we address your questions and feedbacks.
> > >
> > > **IntGRAD MT with 7-cumulative-shot:**
> > >
> > >  To clarify, the n-cumulative shots in the computation scenario are filled to show that the computational cost of Grad MT is compatible with doing series of few-shot translations. This setup was not part of an experiment for scoring purposes. However, if by ‘IntGrad MT with 7-Cumulative-Shot’ you meant ‘path sampling with 7 samples’, please refer to the table below.
> > >
> > > | Setting                  | DE     | ZH     | HI     | KO     | SW     | BN     | MR     |
> > > |--------------------------|--------|--------|--------|--------|--------|--------|--------|
> > > | __Baseline 15 Shot__        | 98.01  | 92.16  | 73.13  | 90.73  | 81.59  | 69.70  | 45.54  |
> > > | **1 Start**                 | 97.85  | 91.36  | 73.89  | 90.69  | 82.18  | 69.84  | 47.36  |
> > > | +Pre                    | 97.87  | 91.90  | 74.80  | 91.31  | 82.44  | 71.84  | 48.53  |
> > > | +Post                   | 98.02  | *92.51*  | 76.44  | 92.29  | 83.96  | 74.16  | 51.71  |
> > > | +Pre & Post             | 97.93  | 92.04  | 76.32  | 91.70  | 83.93  | 73.66  | 50.78  |
> > > | __Path Sampling (3)__       | 97.97  | 91.13  | 78.41  | 92.72  | 83.72  | 74.49  | 52.84  |
> > > | +Pre                    | 98.00  | 91.30  | 79.02  | 92.99  | 84.39  | 76.17  | 55.35  |
> > > | +Post                   | **98.18**  | 92.19  | **80.62**  | **94.16**  | **85.77**  | **78.75**  | **57.95**  |
> > > | +Pre & Post             | *98.13*  | 91.35  | 80.48  | *93.23*  | *84.77*  | *77.64*  | *57.07*  |
> > > | __Path Sampling (7)__       | 97.87  | 91.53  | 74.80  | 90.33  | 82.00  | 69.64  | 46.67  |
> > > | +Pre                    | 97.90  | 91.89  | 75.10  | 90.88  | 82.28  | 71.53  | 47.99  |
> > > | +Post                   | 98.07  | **92.58**  | 77.22  | 92.00  | 83.72  | 73.82  | 51.32  |
> > > | +Pre & Post             | 97.97  | 92.05  | 77.16  | 90.98  | 83.71  | 72.58  | 50.47  |
> > >
> > >
> > > **Mentioning Latency part in another column:**
> > >
> > > We agree that mentioning latency part in another column would enhance clarity. We will incorporate this change in our camera-ready version of tha paper.
> > >
> > > **Missing \toprule:**
> > >
> > > Thank you for pointing out the missing \toprule. We will ensure it is included in our camera-ready version!
> > >
> > > Thank you again for your keen observations and suggestions -- they are crucial to refining our work!

---

### Official Review · Reviewer_3NmE · 2024-11-02

**Soundness:** 2
**Presentation:** 2
**Contribution:** 2
**Rating:** 6
**Confidence:** 4

**Summary:**

The paper introduces IntGrad MT, a technique that enhances the machine translation capabilities of Large Language Models without additional training. It utilizes Sentence Interpolation and Gradual MT to explore the intermediate processes in translation by creating a sequence of translations that progressively increase in complexity. This additional context is used to improve the final translation. Empirical results demonstrate that this method improves translation quality, particularly for low-resource languages such as Hindi, Swahili, Bengali, and Marathi, as measured by xCOMET scores. IntGrad MT offers a practical way to elevate translation performance without relying on extra training data.

**Strengths:**

- The idea of introducing intermediate steps (additional interpolated demonstrations) to guide the LLM in improving translation is interesting. Like CoT and ICL, this represents a training-free translation strategy that achieves enhancement.

- The proposal was tested in experiments, particularly showing significant improvements in low-resource translation scenarios.

- The authors provide insightful analysis and ablation studies in the paper, which are beneficial for understanding the effectiveness of the approach.

**Weaknesses:**

However, there are some weaknesses:

- Lack of strong baselines. The paper does not include some strong multilingual LLMs in the comparison, such as the recent TowerInstruct. Comparisons are only made with its dependent LLMs, which limits the scope of evaluation.

- The left result in Figure 5 does not convincingly support the claim that "the interpolation successfully bridges the two sentences". Question: Could you provide more explanation? The example shown in Appendix B.1 on Page 17 indicates that the sixth interpolation still shows a considerable difference from the end sentence in word-level similarity. What is the COMET score for that sentence?

- Unfortunately, the evaluation was conducted on sentence-level tasks, whereas LLMs excel in document translation. It would be beneficial to see if the interpolation approach is also effective with document-level test data.

- One of my concerns is that there is no analysis or evaluation of computation time during inference. The authors should report this information to make the study more comprehensive.

- The authors claim that their method addresses difficult translation tasks by expanding the areas where the model performs well. However, they did not clearly define what constitutes a 'difficult' level, and there is a lack of an ablation study examining the translation quality among sentences with varying difficulties. Additionally, it is unclear whether it is possible to enhance the model with just a few interpolated sentences for the more challenging cases.

**Questions:**

See the weaknesses

---

> ### Author Response · Authors · 2024-11-27
>
> Dear Reviewer,
> Thank you so much for your thoughtful feedback!
>
>
> ### About the Baselines and Computational Cost
> We deeply appreciate your concerns regarding the baselines and computational cost. To address these, we’ve introduced new baselines and metrics to ensure a more comprehensive and robust evaluation in Appendix F. Additionally, we conducted a detailed analysis of computational costs to offer a clearer understanding of the trade-offs involved in Appendix C. These updates can be found in the general comments section, and we hope they provide the insights you were seeking.
>
>
> ### About the Interpolation
> When analyzing interpolation paths, our focus was to determine whether the sentences along the path progressively became more similar to the end sentence. To investigate this, we evaluated the progress values, checking whether they exceeded 0. As shown on the left side of Figure 5, the majority of progress values are indeed positive, indicating that the interpolated sentences trend closer to the end sentence. The 2D scatter plot on the left side of Figure 5, along with the visualizations in Appendix I, further clarifies this trend.
>
>
> ### Applying IntGrad MT to Document-Level Translation Tasks
> Thank you for suggesting this intriguing idea—it’s genuinely inspiring! However, most existing translation datasets consist of sentence pairs, and the majority of MT research has focused on sentence-to-sentence translation. While your proposal is outside the scope of our current study, we see great potential in this direction and would love to explore it further in the future.
>
>
> ### About the Definition of 'Difficult'
> We define "difficult" sentences as those that a model struggles to translate effectively (i.e. relatively low comet score compared to translations in validation set) in a zero-shot setting. This definition does not rely on a fixed threshold or strict criteria—what might be easy for ChatGPT could pose significant challenges for smaller models like Mistral-Nemo.
> Since this definition is based on intuition and focuses on scenarios where the model translates arbitrary sentences, we did not explicitly analyze the impact of varying difficulty levels. However, we observed larger score improvements in low-resource languages, suggesting that IntGrad MT becomes even more effective as the difficulty of the translation task increases.

---

> ### Author Response · Authors · 2024-12-01
>
> Dear Reviewer 3NmE,
>
> Thank you for sharing your concerns about our paper. We also appreciate your recognition of our contributions, notably our training-free strategy that enhances translation, particularly for low-resource languages, and our detailed ablation studies which lay groundwork for future research.
>
> During the discussion period, we have diligently addressed your concerns, particularly those related to weak baselines and high computational costs.
>
> To provide a broader and more thorough analysis, we have added new metrics (BLEURT and BLEU), new baselines (15-shot, 50-shot, NLLB, TowerInstruct), and an analysis of computational costs.
>
> Additionally, we have incorporated new test results from path sampling and path truncation experiments, aimed at reducing the costs of gradual MT. Notably, path sampling has enabled us to enhance performance while cutting computational costs by half.
>
> As the discussion period is nearing its conclusion, we hope you have had a chance to review our revised paper. We warmly welcome any further feedback or questions you may have.
>
> Thank you.

---

> ### Comment · Reviewer_3NmE · 2024-12-03
>
> Thank you for your efforts in addressing some of my concerns. I hope the authors can integrate these changes into the final version of their paper. I have raised my score accordingly.

---

> > ### Author Response · Authors · 2024-12-03
> >
> > Dear Reviewer,
> >
> > Thank you very much for reviewing our response and updating your scores. We greatly appreciate your feedback and will do our best to incorporate the changes into the final version.
> >
> > Thank you once again for your time and consideration.

---

### Official Review · Reviewer_gKyV · 2024-11-11

**Soundness:** 1
**Presentation:** 3
**Contribution:** 1
**Rating:** 3
**Confidence:** 4

**Summary:**

This paper assumes that a source sentence (called end sentence) has paired sentences (called start sentences) that are easy to translate or have gold translations and these paired sentences with corresponding translations can be used as translation samples in order to prompt the model to translate the source sentences. To use this assumption, the authors presented a multi-step prompting framework:

- Step 0:  use an existing dev set to create a start sentence pool
- Step 1:   select start sentences from the pool based on similarities.
- Step 2:   calculate the interpolation sentences between the source sentence (or end sentence) and the paired start sentences and translate all interpolation, start sentences, and the end sentence.
- Step 3: Chain all sentences and translations into multiple prompts according to the start sentences.
 - Step 4: select the final translation from the results of these prompts.

Experiments are conducted on FLORES-200 for 7 languages.

**Strengths:**

1. The idea is interesting.
2. The paper is clear. I can easily follow the paper.
3. The authors analyze each step in the ablation study.

**Weaknesses:**

1. Evaluation is not fair. The authors use xCOMET to select translations and evaluate the final results on the test set. The final results are biased to xCOMET or COMEwiki scores. The authors have to provide scareBLUE scores to justify the effectiveness. Meanwhile, since you use all dev sets as your start sentences pool, it is not fair to other models. How do you select 3 examples for your 3-shot translation? What is your selection method ? Can you try an experiment where we select 3 similar sentences from dev set with the source sentences as the 3 examples?

2. Efficiency is poor. The method required too many prompts and tokens to finish the translation task. For example, in Table 2, the model requires 3 prompts that generate multiple sentences for translation. Can you provide a study of efficiency?

3. Experiments are limited. The authors claim the framework works for low-resource languages. However, only limited experiments are conducted in Table 1.

**Questions:**

1. Refers to Weaknesses.
2. How many interoplation sentences generated on average for one translation?

---

> ### Author Response · Authors · 2024-11-27
>
> Dear Reviewer,
> Thank you so much for sharing your concerns about our paper. Your detailed feedback has been incredibly valuable, and we’re truly grateful for the opportunity to improve our work.
>
> In response, we’ve made several updates: new baselines, presentations of  additional metrics, and a thorough analysis of computational costs.(Please refer to Appendix C. and Appendix F.)  And after initial submission, we came up with new approaches that can reduce the computational burden, including the idea of shortening the interpolation path used in GradMT  through sampling. The comparison between the previous approach and the new one demonstrates that the new approach reduces the time cost by more than half. Please refer to Table 7 in the paper and the general response. Please check out the revised manuscript and general comments—we hope these updates effectively address your concerns!
>
> That said, we’d like to take a moment to clarify a few key points:
>
> First, the few-shot examples provided to the baselines were directly taken from the start pool used in IntGrad MT. The method for retrieving similar sentences from the start pool employed SBERT similarity, which is the same approach we used for selecting start sentences in IntGrad MT. In essence, the way we incorporated few-shot examples into the baselines mirrors the method you proposed.
>
> Second, it is important to highlight that the models used for evaluation and for IntGrad MT were different. We used xCOMET, with reference for evaluation, and CometKiwi, without reference for IntGrad MT. Since CometKiwi is a reference-free model, there was no performance boost due to peeking at test results—it was a clean, unbiased process. Of course, the COMET family of models might share some underlying architecture or training methodologies, which could introduce minor implicit biases. However, to mitigate such concerns, we presented results using different evaluation metrics, such as MetricX, and complemented these with additional metrics like BLEURT and BLEU.
>
> Finally, to answer your question: the average number of sentences in the interpolation path is about 7.
>
> Thank you once again for your insightful comments and questions. We’re confident these updates bring us closer to meeting your expectations and are excited to hear your thoughts on the revised paper!

---

> ### Author Response · Authors · 2024-12-01
>
> Dear Reviewer gKyV,
>
> We would like to gently remind you to take a look at the revised version of our paper and the new experiments we've included in response to your feedback. We've added some updates that we think you'll find valuable:
>
> - We've incorporated new metrics such as BLEURT and BLEU.
> - We've introduced additional baselines including 15-shot, 50-shot, NLLB, and TowerInstruct.
> - We've conducted a detailed analysis of computational costs.
>
> Furthermore, we've implemented a new experiment with path sampling and path truncation to decrease the computational demands of Gradual MT. I'm pleased to share that these changes have not only halved our computation times but have also significantly enhanced our performance!
>
> Please take a moment to review the general comments section for more important updates.
>
> Thank you very much for your time and attention.

---

> ### Comment · Reviewer_gKyV · 2024-12-02
> **Sorry for late reply**
>
> Thanks for your additional experiments.
>
> For two unbiased scores in Tables 9 and 10,  we can see the gain is marginal, considering lots of computing required.
>
> Another point in your new setting is weird to me. You have found chaining too many examples into the prompt is not promising. Why do not try a small number, e.g., 3 or 5?
>
> Again, using dev for your prompts is not a good idea. Usually, dev and test set are coming from the same distribution. It is likely to fail when you apply for real-word applications. For example, your input is far away from any example in the dev set.

---

> ### Author Response · Authors · 2024-12-02
>
> > For two unbiased scores in Tables 9 and 10, we can see the gain is marginal, considering lots of computing required.
>
> I respectfully disagree that MQM results are marginal improvements. IntGradMT on GPT 3.5  reduces MQM on HI (19%),  KO (12%), SW (9%),  BN(26%), and MR (40%); on 70B reduces MQM (error rate) by DE (12%), HI (25%), and similar trends are shown in Mistral Nemo as well. These statistics are compared against the best performance in baselines (0-, 15-, 50- shot). Similarly, we do not think BLEURT result in Table 10 is marginal. See similar pairs GPT3.5 with HI, SW, BN, MR (+1.4-+2.6), Llam70B (+0.8-+2.6), etc.
>
> We agree that MQM improvement is sometimes marginal on high resources, but we respectfully disagree that all results in Table 9 are marginal improvements. We also want to highlight that Tables 9 and 10 do not hold our best results from 'Path Sampling'. As displayed in Table 4, Path Sampling improves performance for most target languages, however, during the response period, we were unable to change every table to reflect this result.
>
> > Again, using dev for your prompts is not a good idea. Usually, dev and test set are coming from the same distribution. It is likely to fail when you apply for real-word applications. For example, your input is far away from any example in the dev set.
>
> Your comment can result in an interesting experiment. I generally agree with your point; however, I want to clarify that our baselines are also using the same dataset. There is no real advantage given to just us.
>
> Also, imagine this scenario where unlabeled texts are available. In such cases, we believe that we can create such starting points on the fly as well by retrieving similar sentences to the current sentence that one wishes to translate and creating many translations to choose the one that has the highest unreferenced quality estimator (such as Cometkiwi) as the starting point. We cannot think of other methods that can improve its performance just given the model and its translations.
>
> Further, We believe sometimes translation isn't all about speed as translation is required in many applications, including the actual transfer of knowledge from one language to another, such as whole-book translation. In such cases, it would be beneficial to have more accurate results than having faster yet less accurate results.
>
> > Another point in your new setting is weird to me. You have found chaining too many examples into the prompt is not promising. Why do not try a small number, e.g., 3 or 5?
>
> If you meant putting many examples in the prompt is detrimental, that is not necessarily true. If you see 'Path Truncation' v.s. Default IntGradMT in Table 4, the results are quite similar where the Default setting excels more in lower-resource settings and 'Path truncation' performs slightly better in higher-resource settings. In the path, our max number of few-shot examples would be around 10 or so (except some outliers), and this amount of prompt do not show a negative effect based on our experiments. For example, in baseline experiments with 3-, 5-, and 15- shots 50-shots, we actually get the best baselines at 15-shots most of the time.
>
> If you were referring to Path Sampling instead, then I want to clarify that we do not obtain better results by shortening the prompt itself but by sampling the interpolated path. We select start, middle, and target in the Path Sampling, which shows notable improvements. We believe this is due to the noises in the interpolated path, as not every step in the interpolation is meaningful. If you suggested sampling more for points from the path, we did not have enough bandwidth to conduct such an experiment. We would be happy to perform such an ablation study in the updated version of the paper when we get a chance.

---

### Author Response · Authors · 2024-11-25
**General response to the reviews**

We sincerely thank the reviewers for their thoughtful and valuable comments on our work. Based on their feedback, we have updated our paper, with the changes highlighted in blue.

Below, we address the main concerns and questions raised regarding our work:



## Regarding Baselines
The reviewers noted that the baselines in our study appeared relatively weak, especially considering the computational cost of our approach. We acknowledge this concern and have incorporated additional baselines, including 15-shot and 50-shot translations. Also, we have added TowerInstruct and NLLB for broader comparison.

The results indicate that increasing the number of shots does not necessarily enhance model performance; in fact, providing 50-shot translations led to a decline in translation quality. This suggests that increasing the number of shots is not a reliable strategy for further improving the performance of LLMs. Additionally, the results demonstrate that our approach outperforms TowerInstruct and exceeds NLLB in both high- and mid-resource languages.


We also wish to emphasize that our method focuses not only on performance but also on generalizability. By testing our approach across various models, we have shown that IntGrad MT can be applied to off-the-shelf LLMs with reasonable translation capabilities.

__Minor note:__ One of the related works we decided to omit is Chain-of-Dictionary. We were unable to reproduce the results reported in the original paper. Furthermore, its performance was significantly lower than ours, so we decided not to include it as a baseline.



## Metrics
The reviewers also noted that our work relies heavily on the COMET metrics.

To address this concern, we wanted to clarify

(1) that we have presented other metric such as MetricX and

(2) that we use COMET unreferenced for IntGradMT and COMET referenced for evaluation.

On top of this, in this response cycle, we also report BLEU and BLEURT scores to provide a more comprehensive evaluation.

Across all LLMs and all embedding-based metrics (e.g., xComet, CometKiwi, MetricX, and BLEURT), our approach demonstrates superior performance compared to previous methods. For BLEU score, we generally observed marginal improvements in BLEU scores across all LLMs, with one exception of  Mistral-Nemo.  Mistral-Nemo  had a lower BLEU score but a higher BLEURT score (or other embedding-based metrics). A closer human inspection of Mistral-Nemo results (for a language we can read and speak) confirmed that our approach produced more fluent translations, which n-gram-based metrics like BLEU fail to adequately capture.

While we understand the reviewers' concerns regarding our reliance on QE metrics, we respectfully suggest that these metrics -- particularly embedding-based ones like xCOMET and MetricX -- are well-suited for evaluating translation quality. Unlike traditional metrics such as BLEU, which primarily measure surface-level similarity, QE metrics capture more nuanced aspects of fluency and adequacy in translation. This alignment is evident in our findings, where higher QE metric scores consistently correlated with observed improvements in translation quality, even when BLEU scores showed only marginal gains. We hope this clarifies our choice and provides additional context for the evaluation methodology.



## Computatational overhead
One of the pointed weaknesses of our approach is the computational overhead caused by sentence interpolation and Gradual MT. Compared to our baselines, our approach is indeed slower. To clarify this, we systematically analyzed the IntGrad MT process and calculated the expected time required to translate a single sentence for each scenario covered in our paper. As shown in the Table 5 in Appendix C (or the first table below), the primary bottlenecks are Sentence Interpolation and Gradual MT.(https://openreview.net/forum?id=SmxM4POTBk&noteId=3EYYigvQxq)

To mitigate these issues, we explored two strategies. First, we leveraged reference-free QE models to apply IntGrad MT only to poorly performing examples (pre-filtering), effectively reducing the number of interpolations while maintaining overall performance. Second, we investigated ways to optimize Gradual MT, such as selecting a fixed number of sentences from interpolation paths for Gradual MT. Interestingly, this second approach ___not only reduced computational costs but also resulted in substantial improvements in translation quality.___ Please refer to Table 4 in Section 6.3 for the results, or the last table below (https://openreview.net/forum?id=SmxM4POTBk&noteId=uE3pxr1bNn).

While it is true that our method is slower than previous methods, we believe that the performance improvements outweigh the additional computational costs.

---

> ### Author Response · Authors · 2024-11-25
>
> **Table: The computational cost (execution time and memory) for IntGrad.**
> The computational cost (execution time and memory) for IntGrad. The statistics are calculated translating 10 randomly sampled sentences from the FLORES test set into Hindi using Llama-3.1-8b. \textit{Path sampling} denotes the gradual MT operation using only the first, middle, and last steps of the interpolation path. The pre-filtering and post-filtering steps run the QE model using batches of data. We used a batch size of 8, and running one batch took 49.77 seconds. Since the time of running N sentences can be estimated at 49.77 * (N/8), we denote the time per sentence as 6.22(s).
>
> | Step                  | Time per sentence (s) | GPU Peak Mem (GB) |
> |-----------------------|------------------------|--------------------|
> | Zero-shot baseline    | 3.05                  | 5.8                |
> | Pre-filtering (QE)    | 6.22                  | 43.9               |
> | Interpolation         | 26.17                 | 44.2               |
> | Grad MT               | 19.80                 | 6.4                |
> | Path sampling         | 7.00                 | 6.0                |
> | Post-filtering (QE)   | 6.22                  | 49.9               |
>
>
>
>
> **Table: Estimated time for possible filtering strategies of each method. N: \# of sentences, M: \# of sentences after pre-filtering, M=N*0.45 according to statistics from experiments.**
> | Method                  | Scenario                              | Estimation                                   |
> |-------------------------|---------------------------------------|---------------------------------------------|
> | **IntGrad**             | **All**: Interpolation → Grad MT     | (26.17 + 19.80) · N = **45.97N**           |
> |                         | **Post-Filtering (Delta)**: Baseline (0shot) → QE on baseline results → Interpolation → Grad MT → QE on GradMT results | (3.05 + 6.22 + 26.17 + 19.80 + 6.22) · N = **61.46N** |
> |                         | **Pre-Filtering (Thr.)**: Baseline → QE on baseline results → Interpolation → Grad MT | (3.05 + 6.22) · N + (26.17 + 19.80) · M = **29.96N** |
> |                         | **Pre & Post-Filtering(Thr.+Delta)**: Baseline → QE on baseline results → Interpolation → Grad MT → QE on GradMT results | (3.05 + 6.22) · N + (26.17 + 19.80 + 6.22) · M = **32.76N** |
> | **Path Sampling**       | **All**                              | (26.17 + 7.00) · N = **33.17N**            |
> |                         | **Post-Filtering**                   | (3.05 + 6.22 + 26.17 + 7.00 + 6.22) · N = **48.66N** |
> |                         | **Pre-Filtering**                    | (3.05 + 6.22) · N + (26.17 + 7.00) · M = **24.20N** |
> |                         | **Pre & Post-Filtering**    | (3.05 + 6.22) · N + (26.17 + 7.00 + 6.22) · M = **27.00N** |
> | **50-Shot Baseline**    |                                       | **4.36N**                                  |
> | **15-Shot Baseline**    |                                       | **3.31N**                                  |
> | **7-Cumulative-Shot** |                               | **23.80N**                                 |
> | **2-Cumulative-Shot** |                               | **8.76N**                                  |

---

> ### Author Response · Authors · 2024-11-26
>
> **Table: xCOMET scores of IntGrad MT with different strategies to save computational cost of gradual MT.**
> `Default` denotes the strategy which uses every sentence from the interpolation path.
> `Path truncation` denotes the strategy which uses three recent translations for gradual MT.
> `Path sampling` denotes the strategy which uses the start, middle, and end sentences from the interpolation path for gradual MT.
>
> `Pre` and `Post` denote the output selection strategies, referred to as 'Thr.' and 'Delta' in the original paper.
>
> | Setting              | DE     | ZH     | HI     | KO     | SW     | BN     | MR     |
> |----------------------|--------|--------|--------|--------|--------|--------|--------|
> | **15 shot**          | 98.01  | 92.16  | 73.13  | 90.73  | 81.59  | 69.70  | 45.54  |
> | **Default (1 start)**|        |        |        |        |        |        |        |
> |                      | 97.85  | 91.36  | 73.89  | 90.68  | 82.18  | 69.83  | 47.36  |
> | +Pre                 | 97.87  | 91.89  | 74.80  | 91.30  | 82.43  | 71.84  | 48.53  |
> | +Post                | 98.02  | _92.51_| 76.44  | 92.29  | 83.96  | 74.16  | 51.70  |
> | +Pre&Post            | 97.93  | 92.04  | 76.32  | 91.70  | 83.93  | 73.66  | 50.78  |
> | **Path Truncation**  |        |        |        |        |        |        |        |
> |                      | 97.87  | 91.35  | 74.42  | 90.27  | 82.12  | 69.88  | 46.96  |
> | +Pre                 | 97.91  | 91.87  | 74.94  | 90.77  | 82.42  | 71.61  | 48.57  |
> | +Post                | 98.06  | **92.60** | 77.07  | 91.98  | 83.87  | 73.85  | 51.55  |
> | +Pre&Post            | 97.96  | 92.04  | 76.84  | 90.93  | 83.86  | 73.18  | 50.94  |
> | **Path Sampling**    |        |        |        |        |        |        |        |
> |                      | 97.97  | 91.13  | 78.42  | 92.72  | 83.73  | 74.49  | 52.84  |
> | +Pre                 | 98.00  | 91.30  | 79.02  | 92.99  | 84.39  | 76.17  | 55.35  |
> | +Post                | **98.18** | 92.19  | **80.62** | **94.16** | **85.77** | **78.75** | **57.95** |
> | +Pre&Post            | _98.13_| 91.35  | _80.48_ | _93.23_ | _84.77_ | _77.64_ | _57.07_ |

---

### Author Response · Authors · 2024-12-02
**Discussion Reminder (Deadline < 48 hrs)**

We sincerely thank all reviewers for their valuable and insightful feedback. **As the discussion period is nearing its end, we would like to kindly remind reviewers to acknowledge our responses and provide any additional comments they may have.**


Below, we provide **one-line summarization of the changes** made to our paper. For more details, please view individual response and previously written general responses.

---
### **1. Addition of New Baselines**
Following the reviewer’s suggestions, we have included **new baselines such as 15-shot, 50-shot, NLLB, and TowerInstruct and show that IntGradMT outperforms all of them.**


### **2. Inclusion of New Metrics**
To reduce reliance on the COMET family of metrics, we have introduced BLEURT and BLEU as **additional evaluation metrics** on top of existing results with MetricX. In most of cases, IntGradMT outperforms baselines.


### **3. Analysis of Computational Costs**
To provide readers with a clear understanding of the trade-off between computational costs and performance, we have conducted an **in-depth computational cost analysis** and organized a table for it in the general response.


### **4. New Method for Reducing Computational Overhead**
We explored **new strategies to mitigate high computational costs** in Gradual MT: **path truncation and path sampling.** With the ‘path sampling’ approach, we observed **improved performance while reducing the cost by half.**


### **5. Improved Clarity**
To improve the clarity of the IntGrad MT framework, we **revised the terminology** by replacing “thr.” &rarr; “pre-filtering” and “delta” &rarr; “post-filtering.” Additionally, we have enhanced the readability and clarity of Section 3.

---
To the best of our knowledge, **we are the first to propose a methodology that utilizes “Sentence Interpolation” to improve sentence generation tasks.** We believe our research represents a novel approach to problem-solving, as it creates a ‘path of problems’ for any LLM to follow, prompting it to **expand its own capabilities without additional training.** IntGradMT is not only innovative but also effective in enhancing performance. We hope the reviewers will recognize the strengths and contributions of our proposed method.

---

### Meta-Review · Area_Chair_LkFG · 2024-12-19

**Metareview:**

The paper introduces a novel method called IntGrad MT to improve the machine translation capabilities of Large Language Models (LLMs) without requiring additional training. This method is effective for low-resource languages and involves two key techniques: 1）Sentence Interpolation: Begins with an easy-to-translate seed sentence and incrementally modifies it until reaching the source sentence. This helps in creating a sequence of translations that gradually increase in complexity. 2）Gradual MT: Utilizes the model's translations as few-shot examples for translating subsequent sentences, thereby improving translation quality through iterative refinement.

The paper is well-written and easy to follow. However, as highlighted by the reviewers, the computational costs and the marginal improvements in BLEU scores reduce the practicality of the method.

**Additional Comments On Reviewer Discussion:**

As the authors list in one-line summarization of the changes: 1) addition of new baselines 2) inclusion of new metrics 3) analysis of computational costs 4) new method for reducing computational overhead 5) improved clarity.

However, the responses can not dispel the Reviewer gKyV and efMc doubts on the computational costs and marginal improvements. Personally, I favor the traditional approach due to its superior performance (as listed in Table 10 and Table 11) and lower computational cost.

---

### Decision · Program_Chairs · 2025-01-22

Reject